



# Cross validations of the Aeolus aerosol products and new developments with airborne high spectral resolution lidar measurements above the Tropical Atlantic during JATAC

Dimitri Trapon[1], Holger Baars[1], Athena Floutsi[1], Sebastian Bley[1], Adrien Lacour[2,3], Thomas Flament[2,4], Alain Dabas[2], Amin R. Nehrir[5], Frithjof Ehlers[6,7], and Dorit Huber[8]

[1]Leibniz Institute for Tropospheric Research (TROPOS), Leipzig, Germany
[2]CNRM, Université de Toulouse, Météo-France, CNRS, Toulouse, France
[3]Magellium, Ramonville-Saint-Agne, Occitanie, France
[4]EUMETSAT, Darmstadt, Germany
[5]NASA Langley Research Center, Hampton, VA, USA
[6]ESA/ESTEC, Keplerlaan, Noordwijk, The Netherlands
[7]Delft University of Technology, Delft, The Netherlands
[8]DoRIT, Germany

**Correspondence:** Dimitri Trapon (dimitri.trapon@tropos.de)

**Abstract.** The Joint Aeolus Tropical Atlantic Campaign (JATAC) conducted 2022 in Cabo Verde has provided quantitative lidar measurements, in particular from the NASA Langley High Altitude Lidar Observatory (HALO) on-board DC-8 aircraft, for process level understanding of tropical dynamics, as well as for satellite validation. For the first time, optical properties of particles (i.e., backscatter, extinction, attenuated backscatter, and depolarization coefficients) have been measured for ex-

tended tropospheric sections collocated with the Aeolus satellite overpasses with limited geolocation and time offsets. This has contributed to the evaluation of the Aeolus Level-2A (L2A) aerosol optical properties product. In addition, localized aerosol profiles were measured by the ground-based multiwavelength Raman polarization and water-vapor lidar Polly[XT].

    With this study, we assess the quality of the Aeolus L2A product retrieved with the Standard Correct Algorithm (SCA) and the Maximum Likelihood Estimation (MLE) as part of the September 2022 dataset reprocessed with the L2A processor

version 16. The focus is given to the 355 nm aerosol retrievals given at finer horizontal resolution, i.e., so-called Aeolus measurement level of $\approx 18$ km. They are compared to the 532 nm HALO airborne profiles which are converted to 355 nm using the backscatter Ångström exponent. HALO and Polly[XT] polarization lidars also provide insights about the L2A algorithms limitations when looking at non-spherical particles such as Saharan dust. Even though having no cross-polarized component the Aeolus measurements can be corrected using collocated observations with such instruments that include both co-polarized

and cross-polarized components of the backscattered light. Moreover the cross with independent lidar measurements allows to estimate lower limits for Aeolus backscatter detection.



## 1  Introduction

The Aeolus Level-2A (L2A) Aerosol Optical Properties Product (Flament et al., 2021) processed from the first UltraViolet (UV) Doppler lidar in space ALADIN (Atmospheric LAser Doppler INstrument) has been gradually improved since its first
implementation following the launch of the mission in 2018. Initially referred to as a spin-off product the validity of its independent retrieval of the extinction and backscatter coefficients for particles was demonstrated using independent datasets for validation (Abril-Gago et al., 2022; Gkikas et al., 2023) and scientific applications (Baars et al., 2021; Khaykin et al., 2022).

In particular, the use of aerosol retrievals from the so-called Standard Correct Algorithm (SCA) developed at Institut Pierre Simon Laplace (IPSL) and Meteo France, revealed limitations (Baars et al., 2020). This first algorithm corresponds to a direct
inversion of the lidar equations without a-priori conditions and is highly affected by the noise (Flament et al., 2021). In addition to the limited amount of vertical range bins, i.e., up to 24 from 250 m to 2 km thickness and reaching from the ground to a maximum of 30 km altitude, the primary mission of Aeolus was to measure winds, hence the will to focus on aerosol-free regions of the atmosphere with atmospheric signal by molecules from the Rayleigh channel. Because of the attenuation of the overlying molecular atmosphere the Signal-to-Noise Ratio (SNR) decreases with altitude, the smaller values corresponding to
the lowest range bins.

Every version of the L2A processor was coming typically every six months during the operational phase of the Aeolus mission. Together with the technical documentation, each processor version forms an L2A product level labelled as Baseline. A new algorithm was developed using physically constrained optimal estimation to compensate the noisy signals, especially when deriving the extinction coefficients. This new algorithm, called Maximum Likelihood Estimation (Ehlers et al., 2022),
was implemented through L2A processor labelled Baseline 15 by September 2022. The MLE was initially processed on the coarser Aeolus horizontal resolution, i.e., referred as Basic Repeat Cycle (BRC) or observation, and resulting from a signal accumulation over ≈ 90 km, to be aligned with the SCA algorithm. Later, it was decided to provide the MLE at a sub-BRC level with higher horizontal resolution. These retrievals are labelled MLEsub within the L2A product Baseline 16. The L2A algorithms SCA and MLE were maintained and improved at Meteo France until Baseline 15, the activities being transferred to
the Leibniz Institute for Tropospheric Research (TROPOS) by Baseline 16.

The Aeolus dataset corresponding to the 2022 Joint Atlantic Tropical Campaign (JATAC) have been processed with Baseline 16 as part of the 4th reprocessing campaign activities conducted by the Aeolus Data, Innovation and Science Cluster (DISC). A summary of quality of Aeolus data products can be downloaded at https://earth.esa.int/eogateway/documents/d/earth-online/ aeolus-summary-reprocessing-4-fm-b-disc-2024-04-30. The reprocessed dataset allows to assess the performance of both
SCA and newly implemented MLEsub algorithms using independent lidar measurements. The present study focuses on cross sections of the atmosphere up to ≈ 10 km altitude observed by the High Altitude Lidar Observatory (HALO) onboard NASA's DC-8 remote sensing aircraft (Nehrir et al., 2017, 2018; Bedka et al., 2021; Carroll et al., 2022) flying towards north and collocated with Aeolus ascending overpasses above the east Atlantic ocean. The limited geolocation offset (i.e., less than 6 km) and time offset (i.e., less than an hour) of these flights with Aeolus offer a unique opportunity to compare both Aeolus
ALADIN and DC-8 HALO measurements of aerosol optical properties with very close collocation. In addition, direct profiles




above Mindelo, Cabo Verde are selected to be compared with aerosol retrievals from ground-based multiwavelength Raman polarization and water-vapor lidar Polly[XT] (Engelmann et al., 2016; Baars et al., 2016). Combining measurement from satellite, aircraft-based and ground-based instruments allows to get a complete picture of tropospheric aerosols. Independent retrievals of aerosol depolarization ratio provided by HALO and Polly[XT] also help to understand how the cross-polarized component of the light missed by Aeolus impacts the L2A aerosol retrievals (Gkikas et al., 2023). The Cabo Verde region is indeed known to be affected by highly depolarizing particles such as Saharan dust (Haarig et al., 2022; Rittmeister et al., 2017). The HALO and Polly[XT] observations will support further refinement of Aeolus products and may complement the classification database for dominant aerosol type.

It is also important to point that the Aeolus mission was part of the ESA Earth-Observation Programme as a demonstrator for wind lidar technology. Therefore, in addition to the proof of concept for a first UV Doppler lidar in space (Dabas et al., 2008), the cross-comparison of Aeolus L2A data with independent aerosols retrievals are anticipating the next missions such as EarthCARE (Illingworth et al., 2015) and Aeolus-2. Moreover, the quality assessment of Aeolus aerosol product has been mainly achieved with localized ground-based measurement (Abril-Gago et al., 2022; Gkikas et al., 2023; Paschou et al., 2022) during the operational phase until successful re-entry in the earth atmosphere during summer 2023. The present study offers a unique opportunity to analyse longer atmospheric sections up to 626 km long collocated with aircraft observation in various atmospheric conditions.

In this paper the Aeolus L2A algorithms SCA and MLE and the JATAC campaign are introduced. Then, the NASA HALO lidar observations performed during the Convective Processes Experiment - Cabo Verde (CPEX-CV) airborne field campaign (Nowottnick et al., 2024) are presented. The wavelength conversion of the HALO measurements and re-gridding to match Aeolus sampling is described followed by a results section firstly showing direct profiles above Mindelo, Cabo Verde observed by Aeolus and HALO with ground-based lidar Polly[XT]. Then the cross-comparison of the tropospheric sections between Aeolus and re-gridded HALO is described. A conclusion section is enclosing the paper, summarizing the L2A algorithm performance and sharing recommendations for future applications. This will be appreciated by the Atmospheric science community as Aeolus data are gradually disseminated to public since 2021, encouraging scientific studies using the dataset.

## 2 Lidar measurements above the Atlantic in context of the Aeolus Tropical Campaign 2022

### 2.1 Aeolus L2A aerosol product: SCA and MLE algorithms concept

The present paper focuses on Aeolus L2A aerosol and clouds optical properties product retrieved with the SCA and MLE algorithms.

**Standard Correct Algorithm**. The SCA algorithm corresponds to a direct inversion of the lidar equations from the accumulated Rayleigh and Mie signals as measured by Aeolus. Making use of the high spectral resolution lidar (HSRL) capacity of ALADIN the attenuated backscatter coefficients for molecules $\beta_{\mathrm{mol}}^{\mathrm{att}}$ and particles $\beta_{\mathrm{part}}^{\mathrm{att}}$ are resulting of cross-talk correction of the lidar signals from vertical matching between Rayleigh and Mie channels, i.e., calculated from the backscattered signal and the transmission through the atmosphere split up in molecules and particles contributions as described in Eqs. (7) and (8)





of Flament et al. (2021). The particulate backscatter $\beta_{\text{part}}$ and extinction $\alpha_{\text{part}}$ are derived from the cross-talk products and the
molecular backscatter $\beta_{\text{mol}}$ is related to air density, as shown with Eqs. (9), (10) and (14) of Flament et al. (2021). They are
then measured independently without a-priori conditions (Flamant et al., 2008). The range bins are considered homogeneously
populated by particles and the top bin is assumed to contain no aerosol.

    **Maximum Likelihood Estimation**. As the SCA is sensitive to noise, leading to non-physical retrievals in low SNR regions,
a new approach based on optimal estimation was introduced in the L2A processing by Baseline 15. It is referred to as Maximum
Likelihood Estimation (MLE) and consists in a reverse processing with derivation of optical parameters, i.e., the physically
modeled state that agree the most with real signals as illustrated in Eq. (11) of Ehlers et al. (2022). It makes use of the
limited memory quasi-Newton L-BFGS-B algorithm (Zhu et al., 1997). Contrary to the SCA it is a constrained processing with
physical limits assuming vertical collocation between the extinction and the backscatter. The extinction-to-backscatter ratio,
i.e., so-called lidar ratio for particles, can't exceed a range from 2 sr to 200 sr and the extinction coefficient must be positive.

The Aeolus L2A product includes aerosol retrievals from multiple algorithms and varying horizontal resolution. The SCA
coarser sampling corresponds to a signal accumulation of 600 consecutive laser pulses comprised in measurements up to 30,
then averaged over $\approx$ 90 km horizontal to form an observation referred as BRC. The SCA attenuated backscatters are aligned
with measurement level and the other SCA products (e.g., extinction and backscatter coefficients for particles) are aligned with
the BRC level. The BRC level products are affected by uncertainties as the averaging may encompass signal from different
conditions (e.g., broken clouds and non-homogeneous aerosol layer).

    The aerosol retrievals derived with the MLE algorithm were initially aligned with the BRC level, i.e., horizontal resolution
of $\approx$ 90 km. It was decided to provide the MLE at finer resolution, i.e., sub-BRC level with horizontal resolution of $\approx$ 18
km, by Baseline 16. The JATAC September 2022 settings allows to assess both SCA and MLEsub products aligned with this
sub-BRC resolution, also referred as measurement level. Figure 1 illustrates the difference between an Aeolus BRC and the
sub-BRC sampling: the ground track for one BRC of $\approx$ 90 km is shown in Fig. 1a (green) and the five corresponding sub-BRC
profiles of $\approx$ 18 km are illustrated in Fig. 1b.

    The L2A products selected for the study are then aligned with the finer sampling of $\approx$ 18 km with SCA particulate attenuated
backscatter coefficient $\beta_{\text{part}}^{\text{att}}$, MLEsub backscatter coefficient for particles $\beta_{\text{part}}$ and MLEsub extinction coefficient for particles
$\alpha_{\text{part}}$. The Quality Check (QC) flags provided for MLEsub retrievals have been applied. They are based on SNR and error
estimates thresholds described in details in the L2A user guide version 2.2 (Trapon et al., 2022). It must be noted that the
Baseline 16 does not include QC flags for SCA particulate attenuated backscatter which then contains non-physical values
(i.e., negative) because of non-perfect cross-talk correction. It was decided to flag the negative values for the study.



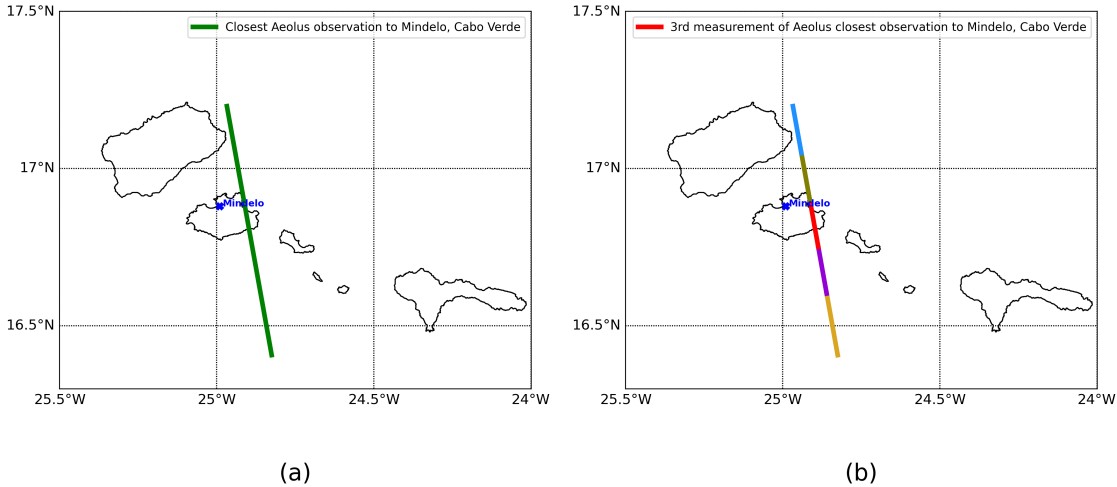

**Figure 1.** Illustration of Aeolus coarser horizontal sampling BRC (observation of $\approx$ 90 km) (a) and corresponding sub-BRC profiles (measurement of $\approx$ 18 km) (b) on 16 September 2022.

## 2.2 The Joint Aeolus Tropical Atlantic Campaign JATAC

JATAC was instigated by the European Space Agency (ESA) in Cabo Verde from June to September 2021 and July to September

2022. The main objective was to provide reference measurement from ground-based observations and atmospheric measurements from balloon and aircraft to validate ESA Aeolus wind and aerosol products retrieved at 355 nm. The ground-based instruments were operated at Mindelo, Cabo Verde. This includes the multiwavelength Raman polarization and water-vapor lidar Polly$^{XT}$ (Engelmann et al., 2016; Baars et al., 2016), the eVe reference polarisation lidar (Paschou et al., 2022), the Airborne Demonstrator for the Direct-Detection Doppler Wind Lidar ALADIN (Lemmerz et al., 2023) onboard DLR Falcon-

20 aircraft, and the High Altitude Lidar Observatory (HALO) onboard NASA's DC-8 remote sensing aircraft (Nehrir et al., 2017, 2018; Bedka et al., 2021; Carroll et al., 2022). Scientific objectives included investigation of Saharan dust interaction and tropical convection (Flamant et al., 2024). Aeolus closest overpasses to Cabo Verde were then selected to be cross analysed with independent lidar measurements. Figure 2 illustrates the Aeolus orbits ground track for September 2022, the NASA DC-8 aircraft flying below the satellite in the same direction toward North for the ascending overpasses only (i.e., Fig. 2 : green (a),

rose (b), yellow (c)).




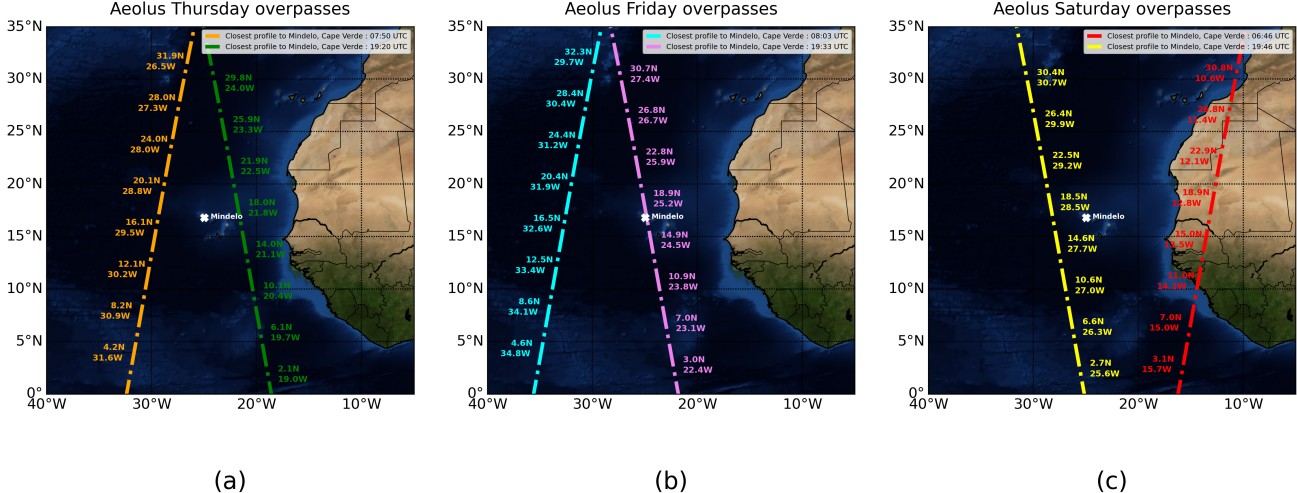

**Figure 2.** Aeolus closest overpasses to Cabo Verde for September 2022.

## 3 NASA's Convective Processes Experiment – Cabo Verde (CPEX-CV)

### 3.1 NASA DC-8 aircraft and flights above Cabo Verde

NASA Convective Process Experiment – Cabo Verde (CPEX-CV) (Zawislak and Zipset, 2023; Nowottnick et al., 2024) is a continuation of the CPEX – Aerosol & Winds (CPEX-AW) campaign conducted in tropical East Atlantic during September 2021. One key objective was to study the atmosphere dynamics in order to interpret space-borne remote measurement of tropospheric aerosols as part of JATAC 2022. CPEX-CV consisted of the deployment of NASA's DC-8 aircraft out of Sal Island, Cabo Verde during summer of 2022. On-board instruments and dropsondes have then be operated to be cross analysed with external aerosol retrievals.

Some flights of the DC-8 have been specifically designed to include a collocated section with the Aeolus overpasses. Three scenes up to 626 km long are presented in this study and correspond to 09 September 2022, 15 September 2022, and 16 September 2022. The scenes 09 September 2022 and 16 September 2022 correspond to Friday ascending overpasses (Fig. 2b, rose). The scene 15 September 2022 corresponds to Thursday ascending overpass (Fig. 2a, green).

### 3.2 NASA Langley High-Altitude Lidar Observatory (HALO) and 532 nm aerosol products

The NASA Langley High-Altitude Lidar Observatory (HALO) on-board the DC-8 aircraft was operated during CPEX-CV which was a part of JATAC 2022. It is an active instrument differential absorption lidar (DIAL) and HSRL with multiple configurations including water vapor DIAL and HSRL, and methane DIAL and HSRL. For CPEX-CV, HALO operated in the water vapor DIAL and HSRL configuration to retrieve water vapor profiles as well as profiles of optical properties of





atmospheric aerosols from multiple wavelength observations (532 nm, 1064 nm). HALO does not emit UV signal at 355 nm such as Aeolus ALADIN and a wavelength conversion is therefore required when carrying out comparisons. The present paper

focuses on HALO retrievals at 532 nm which will be converted to 355 nm, the spectral dependence of optical properties for desert dust being known to be less pronounced between 355 nm and 532 nm than between 355 nm and 1064 nm (Burton et al., 2015; Groß et al., 2013, 2015; Haarig et al., 2017b, 2018; Hofer et al., 2017). Moreover HALO transmits linear polarization whereas Aeolus transmits circular polarization. HALO aerosol extinction and backscatter coefficients, respectively labelled $\alpha_{aer,532}$ and $\beta_{aer,532}$, are taken as input for the cross-comparison with MLEsub retrievals. The attenuated backscatter coefficient

at 532 nm labelled $\beta^{att}_{aer,532}$ is selected for the cross-comparison with SCA retrievals.

The calculation of these 532 nm ready-to-use products is described within the HALO documentation as below and in more detail in Hair et al. (2008).

Aerosol extinction coefficient at 532 nm:

$$\alpha_{aer,532} = -\frac{1}{2}\frac{\delta}{\delta r}ln\left\{\frac{r^2 P^{\parallel}_{mol}}{F\psi\beta_{mol,532}}\right\} - \alpha_{mol,532} \tag{1}$$

With $\beta_{mol,532}$ the molecular backscatter coefficient at 532 nm, $\alpha_{mol,532}$ the molecular extinction coefficient at 532 nm, $r$ the range from the DC-8 aircraft to the measurement bin, $F$ is the transmission of the molecular scattering through the iodine filter, $\psi$ is the transmitter-to-receiver overlap function (which is considered unity beyond 1.0 km range from the aircraft), and $P^{\parallel}_{mol}$ is filtered molecular scattering channel measured through the iodine vapor filter.

Aerosol backscatter coefficient at 532 nm:

$$\beta_{aer,532} = \beta^{\parallel}_{aer,532} + \beta^{\perp}_{aer,532} \tag{2}$$

With $\beta^{\perp}_{aer,532}$ the perpendicular (cross-polarized) aerosol backscatter coefficient at 532 nm and $\beta^{\parallel}_{aer,532}$ the parallel (co-polarized) aerosol backscatter coefficient at 532 nm.

Linear aerosol depolarization ratio at 532 nm:

$$\delta_{lin,532} = \frac{\beta^{\perp}_{aer,532}}{\beta^{\parallel}_{aer,532}} \tag{3}$$

Total scattering ratio at 532 nm:

$$\sigma_{aer,532} = \frac{\beta^{\parallel}_{aer,532} + \beta^{\perp}_{aer,532}}{\beta^{\parallel}_{mol,532} + \beta^{\perp}_{mol,532}} + 1 \tag{4}$$

With $\beta^{\perp}_{mol,532}$ the perpendicular (cross-polarized) molecular backscatter coefficient at 532 nm and $\beta^{\parallel}_{mol,532}$ the parallel (co-polarized) molecular backscatter coefficient at 532 nm.




## 4 Methodology

### 4.1 Conversion of the HALO 532 nm atmospheric products to parallel 355 nm

Aeolus ALADIN transmits circularly polarized light. The conversion of HALO signal to 355 nm then implies to derive the circular aerosol depolarization ratio at 532 nm using equation (A14) from Paschou et al. (2022):

$$\delta_{\text{circ},532} = \frac{2 * \delta_{\text{lin},532}}{1 - \delta_{\text{lin},532}} \tag{5}$$

Aeolus only measures the co-polarized component of the backscattered light, i.e., parallel to the transmitted polarization. Therefore to be fairly compared with Aeolus the HALO parallel-only aerosol backscatter 532 nm coefficient $\beta^{\parallel}_{\text{aer},532}$ is derived from Eqs. (2), (3) and (5):

$$\beta^{\parallel}_{\text{aer},532} = \frac{\beta_{\text{aer},532}}{\delta_{\text{circ},532} + 1} \tag{6}$$

The $\delta_{\text{lin},532}$ and derived $\delta_{\text{circ},532}$ include Not a Number (NaN) values which represent noisy retrievals below the value of zero depolarization. It was decided to replace these NaN with value of zero. This allows to assess the Aeolus algorithm performance even for regions of the atmosphere without depolarizing particles.

It is proposed to convert the HALO 532 nm aerosol backscatter $\beta_{\text{aer},532}$ and extinction coefficients $\alpha_{\text{aer},532}$ to 355 nm using equations (1) and (3) of Ansmann et al. (2002) below with $\lambda1 < \lambda2$, $\gamma_{\beta,\lambda1,\lambda2}$ the backscatter-related Ångström exponent and $\gamma_{\alpha,\lambda1,\lambda2}$ the extinction-related Ångström exponent:

$$\gamma_{\beta,\lambda1,\lambda2} = -\frac{ln[\beta_{\lambda1}/\beta_{\lambda2}]}{ln(\lambda1/\lambda2)} \tag{7}$$

$$\gamma_{\alpha,\lambda1,\lambda2} = -\frac{ln[\alpha_{\lambda1}/\alpha_{\lambda2}]}{ln(\lambda1/\lambda2)} \tag{8}$$

Then the HALO 355 nm parallel aerosol backscatter $\beta^{\parallel}_{\text{aer},355}$ and extinction coefficient $\alpha_{\text{aer},355}$ can be derived from Eqs. (7) and (8):

$$\beta^{\parallel}_{\text{aer},355} = \exp\left[ln(\beta^{\parallel}_{\text{aer},532}) - \gamma_{\beta,355,532}ln\left(\frac{355}{532}\right)\right] \tag{9}$$

$$\alpha_{\text{aer},355} = \exp\left[ln(\alpha_{\text{aer},532}) - \gamma_{\alpha,355,532}ln\left(\frac{355}{532}\right)\right] \tag{10}$$

Backscatter-related Ångström exponent $\gamma_{\beta,355,532}$ and extinction-related Ångström exponent $\gamma_{\alpha,355,532}$ have been measured continuously by Polly[XT] at Mindelo. The profiles of the optical properties are cloud-screened and are quality assured. A Saharan Air Layer (SAL) was frequently observed between 1.5 and 4.0 km height. During September 2022, a median backscatter-related Ångström exponent of –0.03 (–0.55 and 0.76 are the 25th and 75th percentile, respectively) was derived in this height range (SAL) from 97 profiles during nighttime. In the same period and height range, a median extinction-related Ångström exponent





of 0.13 (–0.46 and 0.76 are the 25th and 75th percentile, respectively) was derived from 143 Raman lidar profiles during
nighttime. These values are considered to be representative for September 2022. The close-to-zero estimations of backscatter-
related Ångström exponent and extinction-related Ångström exponent are known to be representative for mineral dust (Floutsi
et al., 2023; Haarig et al., 2022). These mean values have been extended to top altitude of 10 km as the HALO dominant aerosol
type product reveals similar aerosol load below and above 4 km altitude (see Appendix A1, Appendix A2, and Appendix A3)
and because the impact on tropospheric background in aerosol-free regions is assumed to be minor. Similarly, monthly median
values of 0.32 (0.12 - 0.58) for the backscatter-related Ångström exponent (101 nighttime profiles) and 0.31 (0.06 – 0.56) for
the extinction-related Ångström exponent (220 nighttime profiles) have been estimated by Polly$^{XT}$ for the height range of 0.8
– 1.0 km corresponding to the Planetary Boundary Layer (PBL). The values in brackets again correspond to the 25th and 75th
percentile. The same values are applied for the section going from the ground to 1.5 km altitude for the present study.

The HALO 355 nm attenuated backscatter coefficient for particles $\beta_{\text{aer},355}^{\text{att}\parallel}$ can be calculated with Eqs. (9) and (10) as follows:

$$\beta_{\text{aer},355}^{\text{att}\parallel} = \beta_{\text{aer},355}^{\parallel}(R) exp\left[-2 \int\limits_{z_{aircraft}}^{z} \left(\alpha_{\text{mol},355}(y) + \alpha_{\text{aer},355}(y)\right) dy\right] \tag{11}$$

With $R$ the DC-8 aircraft range-to-target, $dy$ the range bin thickness, $z_{aircraft}$ the Global Positioning System (GPS) altitude
of the DC-8 aircraft, $z$ the altitude of mean sea level, $\alpha_{\text{mol},355}$ the molecular extinction coefficient for dry air. The $\alpha_{\text{mol},355}$ below
the DC-8 is derived from pressure $p$(hpa) and temperature $T$(K) measured by HALO. The equation (4.6) of Aeolus Level 2A
Algorithm Theoretical Basis Document (ATBD) (Flamant et al., 2022) shown below has been used for the calculation of
$\alpha_{\text{mol},355}$ using a coefficient of 1.16 which was determined experimentally (Collis and Russell, 1976).

$$\alpha_{\text{mol},355}(y) = 1.16\left(\frac{550}{355}\right)^{4.09} \frac{p(y)}{1013} \frac{288}{T(y)} 10^{-5} \tag{12}$$

The molecular extinction above the DC-8 up to 80 km altitude is calculated using the same equation with pressure and
temperature information from Numerical Weather Prediction (NWP) model from the European Centre for Medium-Range
Weather Forecasts (ECMWF). It has to be noted that the flight altitude of the DC-8 is constant enough for the three collocated
sections selected for the study to neglect variations. A mean value of $5.0^{-4}$ has then been derived and used as a constant for all
scenes. This constant is added to the $\alpha_{\text{mol},355}$ in Eq. (11) when calculating the $\beta_{\text{aer},355}^{\text{att}\parallel}$.

Aeolus MLEsub extinction and backscatter coefficients for particles as SCA particulate attenuated backscatter coefficient
can then be compared with the 355 nm converted HALO atmospheric products. Only the valid bins with positive values are
considered for statistics (i.e., the invalid measurements coded as NaN in one product being ignored on the second dataset and
vice-versa). It was decided to not apply HALO and Aeolus cloud masks to keep as many valid bins as possible. This helps
assessing how well Aeolus observes Saharan dust and mixture of aerosol and cloud, the HALO dominant aerosol type product
being used as ad-hoc classification.





## 4.2 HALO re-gridding onto Aeolus sampling

One of the major advantages of Aeolus is the adjustable vertical sampling with Range Bin Settings (RBS). Specific settings have then been set for the region of interest over the Atlantic covering latitudes 12° N to 22° N and longitudes 19° W to 31° W. The top height bin stops at ≈ 18 km altitude and the following vertical sampling have been applied: 0.5 km from 0 to ≈ 2 km altitude, 0.75 km from ≈ 2 km to ≈ 8 km altitude, 1 km from ≈ 8 km to ≈ 18 km. The horizontal sampling is aligned with ≈ 90 km for the coarser observation level and with ≈ 18 km for finer measurement level used for the comparison with HALO
data.

The HALO 532 nm channel comes with a digital filter of 30 m along the vertical and the atmospheric products are calculated from an over-sampled ≈ 15 m resolution. The data then have 30 m vertical resolution with a 15 m sampling interval. The atmospheric data products are calculated from the 15 m interpolated altitude but some products, e.g., extinction, are calculated from further 300 m vertical average. The HALO data are sampled each ≈ 0.5 seconds, a 10 second average being applied to
the backscatter coefficient along the direction of flight. A ready-to-use quality flagging based on molecular backscatter signal and referred as mask_low was applied. A cloud identification tag and height named cloud_top_height was used and plotted over HALO atmospheric products (Figs. 5a-c, 7a-c, 9a-c).

The resolutions of Aeolus ALADIN and DC-8 HALO instruments are then not aligned. Figure 3b illustrates the Aeolus finer horizontal resolution, i.e., so-called measurement level (solid black lines), superimposed over HALO 532 nm aerosol
backscatter coefficient. Then 225 consecutive HALO profiles covering a collocated section of ≈ 538 km on 16 September 2022 correspond to 6 Aeolus BRC (Fig. 3a) and 30 Aeolus profiles (Fig. 3b). The focus being given to Aeolus performance, the HALO atmospheric products have been re-gridded onto the coarser Aeolus measurement scale. The HALO 355 nm converted aerosol extinction, the parallel-only backscatter and the attenuated backscatter coefficients are then re-gridded onto Aeolus grid by deriving the arithmetic mean per range bin for each profile. The equation below corresponds to the HALO 355 nm
backscatter averaged over the Aeolus grid illustrated in Fig. 3b:

$$\beta_{\text{iAeolus}}(\text{Mm}^{-1}\text{sr}^{-1}) = \frac{\sum \beta_{\text{iHALO}}(\text{Mm}^{-1}\text{sr}^{-1})}{N} \tag{13}$$

With $N$ the amount of HALO bins iHALO included per Aeolus range bin iAeolus.

Invalid HALO measurements that correspond to localized conditions, e.g., below dense clouds or within the PBL, and reported as NaN values in the ready-to-use products are not taken into account. The re-gridding is performed even where
the Aeolus coarser grid encompasses valid and invalid HALO retrievals (e.g., white section of HALO $\beta_{\text{aer},532}$ close to profile 114 [15.7° N - 24.7° W] in Fig. 3b), hence a gap filling. Moreover the atmospheric conditions above the DC-8 aircraft have been analysed using Aeolus aerosol products and only the tropospheric sections with clear sky conditions above the DC-8 are considered for the study. This allows to reject collocated profiles where the Aeolus signal can be expected to be too much attenuated, the cross-comparison with the HALO measurements being therefore less representative.



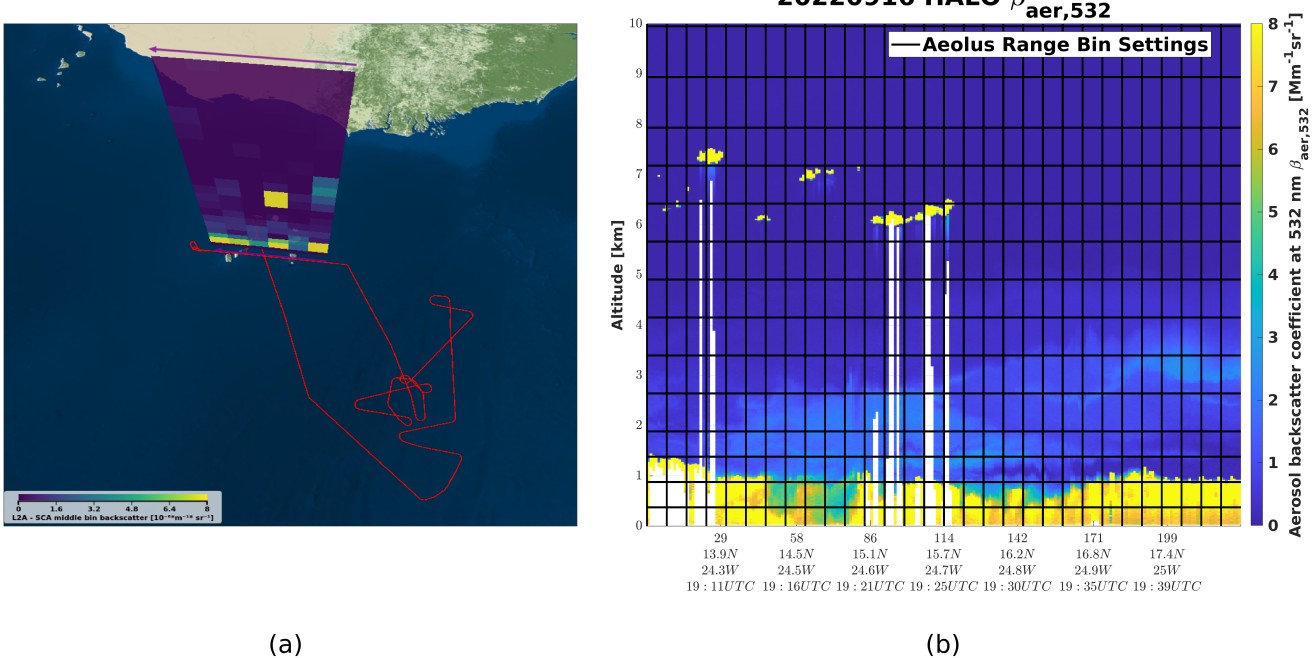

**Figure 3.** Aeolus direct ascending overpass above Cabo Verde through ESA VirES (Downloaded on August 07 2023, Courtesy: https://aeolus.services/) on 16 September 2022 (a) showing the backscatter coefficient for particles derived with SCA algorithm (averaged over consecutive vertical bin and referred as middle bin). The horizontal resolution is aligned with coarser BRC level ≈ 90 km. The top altitude is restricted to 10 km and the range bins are not displayed in scale vertically. The NASA DC-8 flight track is superimposed in red color code and the 6 consecutive BRC level observations of Aeolus orbit file no. 23562 are displayed (a). They correspond to 30 Aeolus measurements given at sub-BRC level ≈ 18 km, and to 225 HALO profiles (b). The HALO aerosol backscatter coefficient at 532 nm $\beta_{aer,532}$ measured for the collocated section on the same day is shown in (b) with superimposed Aeolus Rayleigh sub-BRC grid (solid dark lines).

## 5 Results

### 5.1 Intercomparison of aerosol profiles collocated with ground-based instrument

The HALO atmospheric product converted to 355 nm have been compared to Aeolus for three case studies on 09 September 2022, 15 September 2022, and 16 September 2022. These three case studies correspond to collocated sections of respectively ≈ 626 km, ≈ 537 km and ≈ 538 km long. Figure 4a illustrates how the Aeolus MLEsub $\beta_{part}$ up to 18 km altitude helps assessing the particle-free conditions above the DC-8 flying at ≈ 10.7 km altitude for the 16 September 2022 case. The 2D profiles above Cabo Verde from Aeolus MLEsub and HALO are plotted against ground-based lidar Polly[XT]. Polly[XT] emitting linear polarization (Engelmann et al., 2016), the circular depolarization ratio at 355 nm is derived to recompute the parallel backscatter coefficient at 355 nm with equations similar to Eqs. (5) and (6). Plotting Polly[XT] retrievals in addition to Aeolus and HALO for direct profiles above Mindelo, Cabo Verde allows to validate the wavelength and total to parallel conversions




before crossing Aeolus and HALO with full collocated section. Figure 4b shows how the HALO 532 nm (i.e., dark green) was converted to parallel 355 nm (i.e., dark yellow). The HALO parallel 355 nm profile agrees with Aeolus MLEsub (i.e., violet). Similar assessment can be made focusing on Polly$^{XT}$ 355 nm total (i.e., red) to parallel signal (i.e., blue). The parallel 355 nm profile retrieved with Raman Polly$^{XT}$ indeed agrees very well with MLEsub $\beta_{part}$ from top of Saharan dust layer around 5 km altitude to the in-core layer at 3 km altitude for which MLEsub $\beta_{part}$ of 1.36 Mm$^{-1}$sr$^{-1}$ ± 0.56 Mm$^{-1}$sr$^{-1}$ is measured. The

red curve represents what Aeolus should see if it would have included a polarization channel, then increasing the backscatter signal by a factor ≈ 30 % for the in-core dust layer. When looking at the extinction coefficient for particles, Aeolus MLEsub also agrees with the other instruments even if the Aeolus signal appears to be too much attenuated and therefore flagged below 3 km altitude. One could notice the high errors for MLEsub $\alpha_{part}$. Similar observations can be made with case 09 September 2022 (see Appendix A4), e.g., the Aeolus MLEsub being able to retrieve coherent optical properties in low altitude ≈ 1 km

with high backscatter up to 9.45 Mm$^{-1}$sr$^{-1}$ ± 2.12 Mm$^{-1}$sr$^{-1}$ below dust (see Appendix A4b).

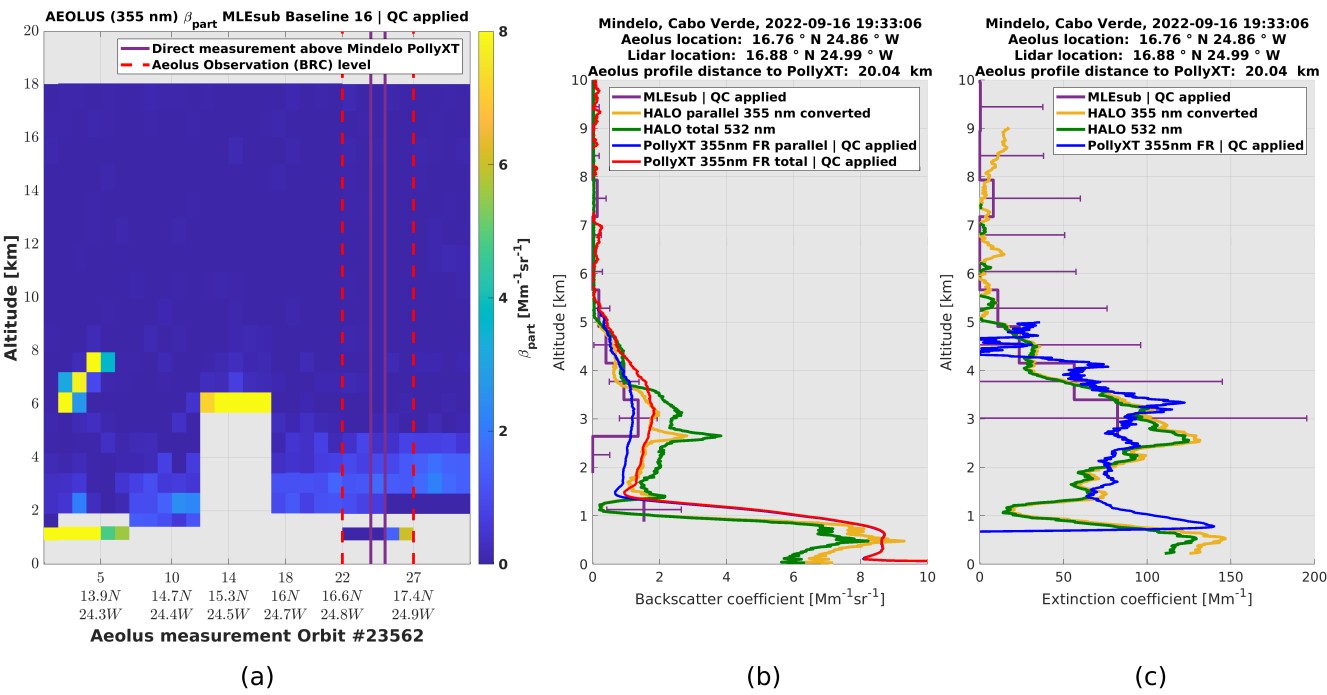

**Figure 4.** Aeolus MLEsub $\beta_{part}$ on 16 September 2022 (a) and cross of 2D profiles above Mindelo, Cabo Verde with HALO and Polly$^{XT}$ for MLEsub $\beta_{part}$ (b) and MLEsub $\alpha_{part}$ (c).

## 5.2  Cross comparison of tropospheric sections observed by Aeolus ALADIN and DC-8 HALO

A linear regression has been applied with the log transformed signals. Working with logarithmic scale helps decreasing the sensibility to outliers which may be expected in low SNR regions. Moreover it offers a better visualization of regimes within two-dimensional histograms. The three indicators R$^2$, slope of the regression line and the Root Mean Square Error (RMSE)





have been calculated with the log transformed signals for each collocated section. The Table 1 below summarizes the scores. The $R^2$ for MLEsub backscatter $\beta_{\mathrm{part}}^{\parallel}$ and MLEsub extinction $\alpha_{\mathrm{part}}$ look solid, i.e., respectively up to 0.83 and 0.76, showing good agreement with HALO retrievals. Interestingly the higher $R^2$ scores correspond to 09 September 2022 for both MLEsub backscatter and SCA particulate attenuated backscatter $\beta_{\mathrm{part}}^{\mathrm{att}\parallel}$, the 15 September 2022 case showing the higher $R^2$ for MLEsub extinction. For each scene the section details are introduced before analysing the cross between HALO converted signals and the Aeolus products.

| | | Sep. 09 | Sep. 15 | Sep. 16 |
|---|---|---|---|---|
| $\beta_{\mathrm{part}}^{\parallel}$ | $R^2$ | 0.83 | 0.75 | 0.77 |
| | Slope | 0.89 | 0.71 | 0.83 |
| | RMSE | 0.58 | 0.45 | 0.60 |
| $\alpha_{\mathrm{part}}$ | $R^2$ | 0.64 | 0.76 | 0.50 |
| | Slope | 1.08 | 1.09 | 0.71 |
| | RMSE | 0.67 | 0.57 | 0.71 |
| $\beta_{\mathrm{part}}^{\mathrm{att}\parallel}$ | $R^2$ | 0.69 | 0.53 | 0.37 |
| | Slope | 0.39 | 0.29 | 0.23 |
| | RMSE | 0.42 | 0.43 | 0.57 |

**Table 1.** Summary table showing the statistics of the linear regression Aeolus MLEsub $\beta_{\mathrm{part}}^{\parallel}$, MLEsub $\alpha_{\mathrm{part}}$, and SCA $\beta_{\mathrm{part}}^{\mathrm{att}\parallel}$ versus HALO 355 nm converted and re-gridded signals. The scores are derived from the log transformed signals.

### 5.2.1 Case study 9 September 2022

**Cross section details**. Figures 5a-c show HALO 355 nm converted (top) and re-gridded (middle), and Aeolus 355 nm (bottom) aerosol retrievals for a collocated section of $\approx 626$ km horizontal from $\approx [13.3° \mathrm{N} - 24.3° \mathrm{W}]$ to $\approx [18.8° \mathrm{N} - 25.3° \mathrm{W}]$. A total of 271 HALO profiles can be re-gridded onto 35 collocated Aeolus measurements. Geolocation offset for the first collocated measurement is equal to $\approx 5.2$ km and initial time offset is equal to $\approx 9$ minutes.

**HALO against Aeolus MLEsub**. HALO cloud_top_height superimposed over parallel 355 nm converted particulate backscatter revealed a marine boundary layer (MBL) up to $\approx 1$ km altitude with values above 8 $\mathrm{Mm}^{-1}\mathrm{sr}^{-1}$ (Fig. 5a top, light yellow color code). Low altitude clouds are dominant for HALO profiles $\approx 103$ [15.3° N - 24.6° W] and last tier of the section above profile $\approx 205$ [18° N - 25° W]. Stratified marine and dusty mix aerosol layers can be seen up to $\approx 4$ km altitude with backscatter from $\approx 3$ to 5 $\mathrm{Mm}^{-1}\mathrm{sr}^{-1}$ (Fig. 5a, light blue to green color codes) as confirmed with HALO dominant aerosol type (see Appendix A1). Aeolus captures well the scene even applying the restrictive quality flags which are removing the lowest range bin below 1 km altitude except for cloudy region below dust in the last tier of the orbit section. The signature of dust particles mixture can indeed be observed on top of the MBL (Fig. 5b bottom, light blue to green pattern with extinction close to $\approx 150$ $\mathrm{Mm}^{-1}$). The signature of the dust layer at $\approx 3$ km altitude for second-last measurement 34 is even more clear. The Aeolus





extinction is also flagged below the dusty mix around $\approx 1.8$ km altitude. This is a confirmation of a signal attenuation with low SNR.

Plotting the 355 nm converted and re-gridded HALO against Aeolus MLEsub for the whole scene reveals $R^2$ scores of 0.83 for backscatter (Fig. 6a) and 0.64 for extinction (Fig. 6b) with respective slope for the regression line of 0.89 and 1.08. The agreement between Aeolus and HALO then appears consistent. Two dominant regimes can be identified using HALO

532 nm circular aerosol depolarization ratio $\delta_{\mathrm{circ,532}}$. The first one is indicated in cyan color code and corresponds to non-depolarizing particles with low backscatter signal (Fig. 6a). The aerosol loading is below a scattering ratio threshold of 0.2 which is required for HALO aerosol classification, hence the Not identified type (Fig. 6d). The second regime comes with dominant yellow color code with circular aerosol depolarization ratio above $\approx 0.3$ (Figs. 6a-b), then mainly attributed to dusty conditions (Freudenthaler et al., 2009). It is confirmed with a dusty mix class (Figs. 6d-e, dark violet). The best agreement

coincides with the second regime connected to highly depolarizing particles. The use of HALO total scattering ratio at 532 nm $\sigma_{\mathrm{aer,532}}$ confirms that the first regime correspond to low scattering (see Appendix A6, cyan color code). This provides further evidence that the first regime is linked to low SNR regions of the atmosphere.

**HALO against Aeolus SCA**. Evidence of the MBL is visible with SCA particulate attenuated backscatter $\beta_{\mathrm{part}}^{\mathrm{att}}$ (Fig. 5c, bottom) and the values look coherent even for the low range bins (i.e., up to $\approx 3^{-3}$ km$^{-1}$sr$^{-1}$). The top of the MBL is captured

for the overall section. The signatures of dust above $\approx 2$ km altitude appear less pronounced compared to HALO 355 nm converted signal (Fig. 5c, light blue color code) with values below $\approx 2^{-3}$ km$^{-1}$sr$^{-1}$ which are hardly distinguishable from background noise. The scatter diagram shows an underestimation of the particulate attenuated backscatter for the second regime by SCA despite a solid $R^2$ score of 0.69. The SCA seems to overestimate the backscatter for the first regime in low scattering regions of the atmosphere (Appendix A7, cyan color code). When removing this regime from the analysis using HALO $\sigma_{\mathrm{aer,532}}$

and applying a threshold of 1.1 the $R^2$ and slope respectively increase from 0.69 to 0.72 and from 0.39 to 0.57, the RMSE decreasing from 0.42 to 0.37. A positive impact is also observed for MLEsub extinction (e.g., the RMSE decreasing from 0.67 to 0.39). Interestingly, a third regime with depolarization from $\approx 0.1$ to $\approx 0.30$ and high backscatter can be identified (Fig. 6c, dark blue to red color codes). The HALO dominant aerosol type points to marine aerosols (Fig. 6f, blue color code). The SCA appears to underestimate the signal for this marine regime.





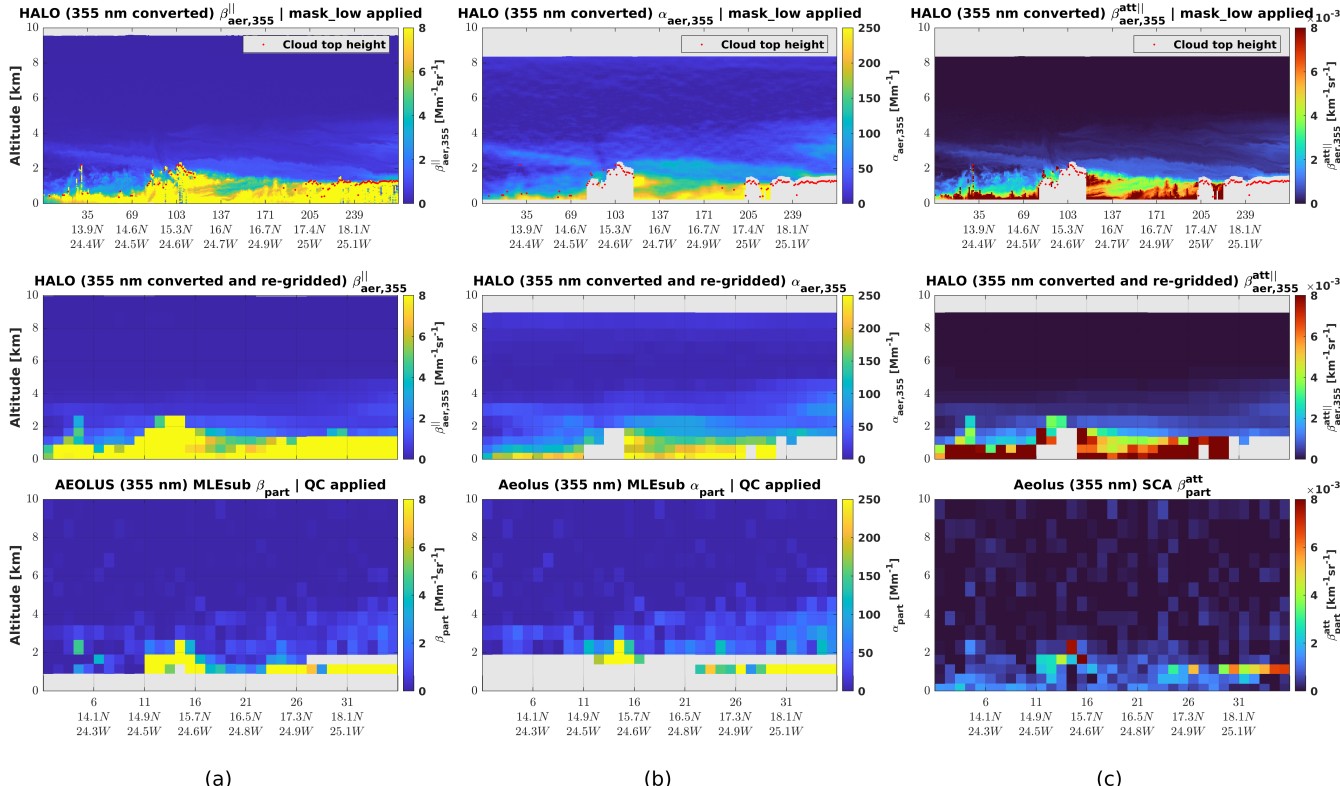

**Figure 5.** Aeolus MLEsub $\beta_{\mathrm{part}}$ (a), MLEsub $\alpha_{\mathrm{part}}$ (b), and SCA $\beta_{\mathrm{part}}^{\mathrm{att}}$ (c) cross section collocated with HALO 355 nm converted products on 09 September 2022.





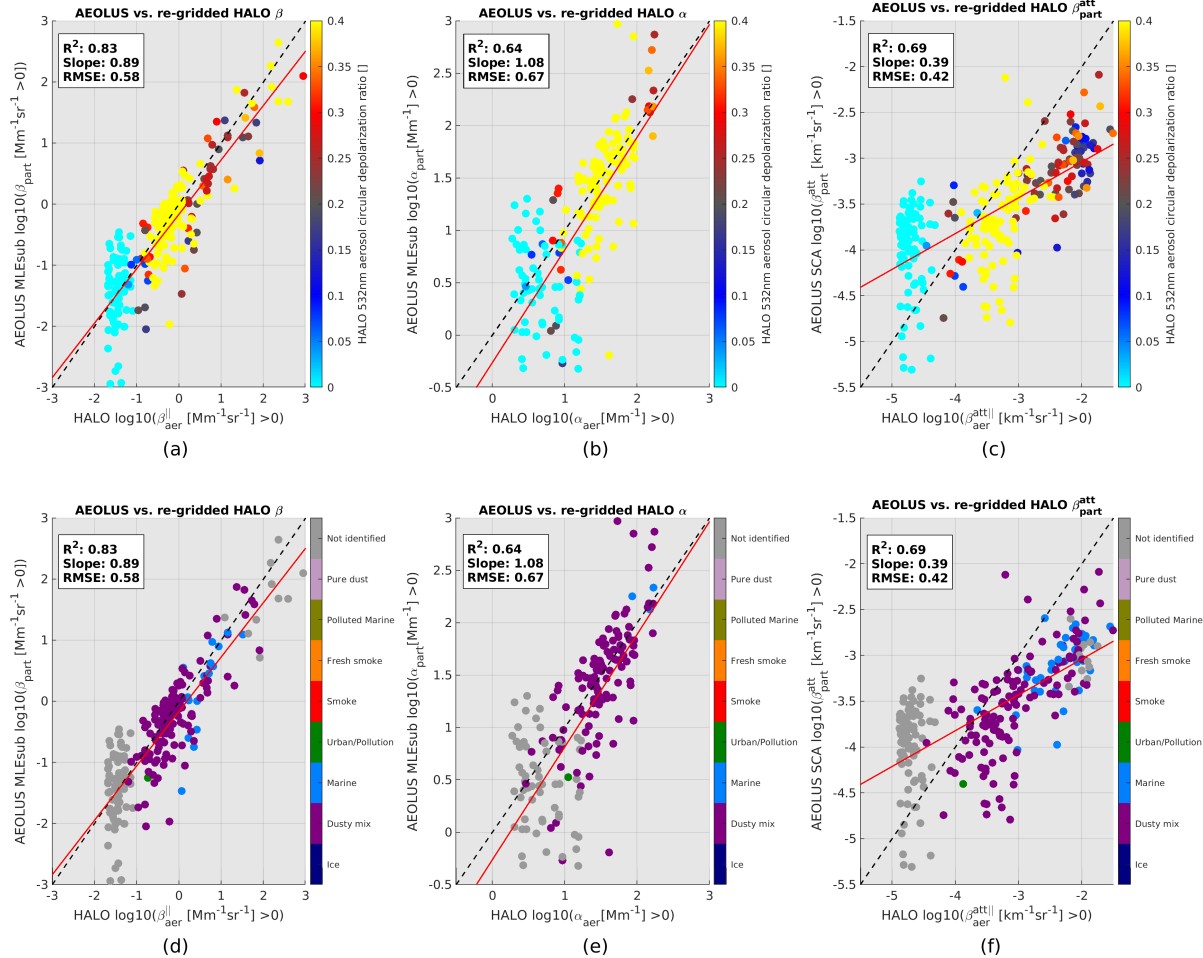

**Figure 6.** Two-dimensional histograms of Aeolus versus HALO 355 nm converted products with HALO $\delta_{\mathrm{circ,532}}$ as colors (a-c) and with HALO dominant aerosol type as colors (d-f) on 09 September 2022. A logarithmic transformation has been applied to the signals.



## 5.2.2 Case study 15 September 2022

**Cross section details**. Figures 7a-c show HALO 355 nm converted (top) and re-gridded (middle), and Aeolus 355 nm (bottom) aerosol retrievals for a collocated section of $\approx 537$ km horizontal from $\approx [15.9°$ N - $21.5°$ W] to $\approx [20.7°$ N - $22.4°$ W]. A total of 218 HALO profiles can be re-gridded onto 30 collocated Aeolus measurements. Geolocation offset for the first collocated measurement is equal to $\approx 3.8$ km and initial time offset is equal to $\approx 13$ minutes.

**HALO against Aeolus MLEsub**. HALO cloud_top_height superimposed over parallel 355 nm converted particulate backscatter shows optically dense MBL up to 0.5 km altitude (Fig. 7a, top). An homogeneous feature, spatially distributed over the scene, with backscatter up to $\approx 5$ Mm$^{-1}$sr$^{-1}$ can be observed around 4 km altitude (light green to yellow color codes) which is typical for Saharan dust layer height (Ansmann et al., 2009). HALO dominant aerosol type confirms the presence of dusty mix (see Appendix A2). The aerosol layer is clearly captured by the MLEsub algorithm, the southern edge of the aerosol layer being particularly visible up to profile 18 [18.7° N - 21.9° W] with both MLEsub backscatter and extinction coefficients (Figs. 7a-b). Both MLEsub products are flagged below this dense and homogeneous aerosol layer. The backscatter diagram show solid agreement between Aeolus and HALO for the dusty mix regime (dark violet color code in Fig. 8d) with R$^2$ score of 0.75. This regime corresponds to a circular aerosol depolarization ratio above $\approx 0.30$ (Fig. 8a, yellow color code). The main deviations correspond to the first regime where aerosol loading is weak and falls below the classification threshold of 0.2 in HALO aerosol scattering ratio at 532 nm $\sigma_{aer,532}$.

**HALO against Aeolus SCA**. The Aeolus SCA algorithm is able to capture well the dusty mix layer at $\approx 4$ km (Fig. 7c bottom, light blue color code) and the MBL below $\approx 0.8$ km. Values of particulate attenuated backscatter up to $\approx 8.0^{-4}$ km$^{-1}$sr$^{-1}$ are measured for the southern edge of the aerosol layer close to profile 5 [16.6° N - 21.6° W]. The scatter diagram shows R$^2$ score of 0.53 and the higher deviations are once more observed for low scattering (see Appendix A7) mostly distributed above 6 km altitude. The better agreement is observed for the dusty mix regime (yellow color code in Fig. 8c). The aerosol layer signature is less clear in the last tier of the scene, i.e. after profile 18 [18.7° N - 21.9° W] with SCA $\beta_{part}^{att}$ below $\approx 2.0^{-4}$ km$^{-1}$sr$^{-1}$. This may be used as a first estimation for lower limit of SCA $\beta_{part}^{att}$ detection for dust particles. Removing the contributions from low scattering below HALO $\sigma_{aer,532}$ of 1.1 has a significant impact on the slope (i.e., increased from 0.29 to 0.48).





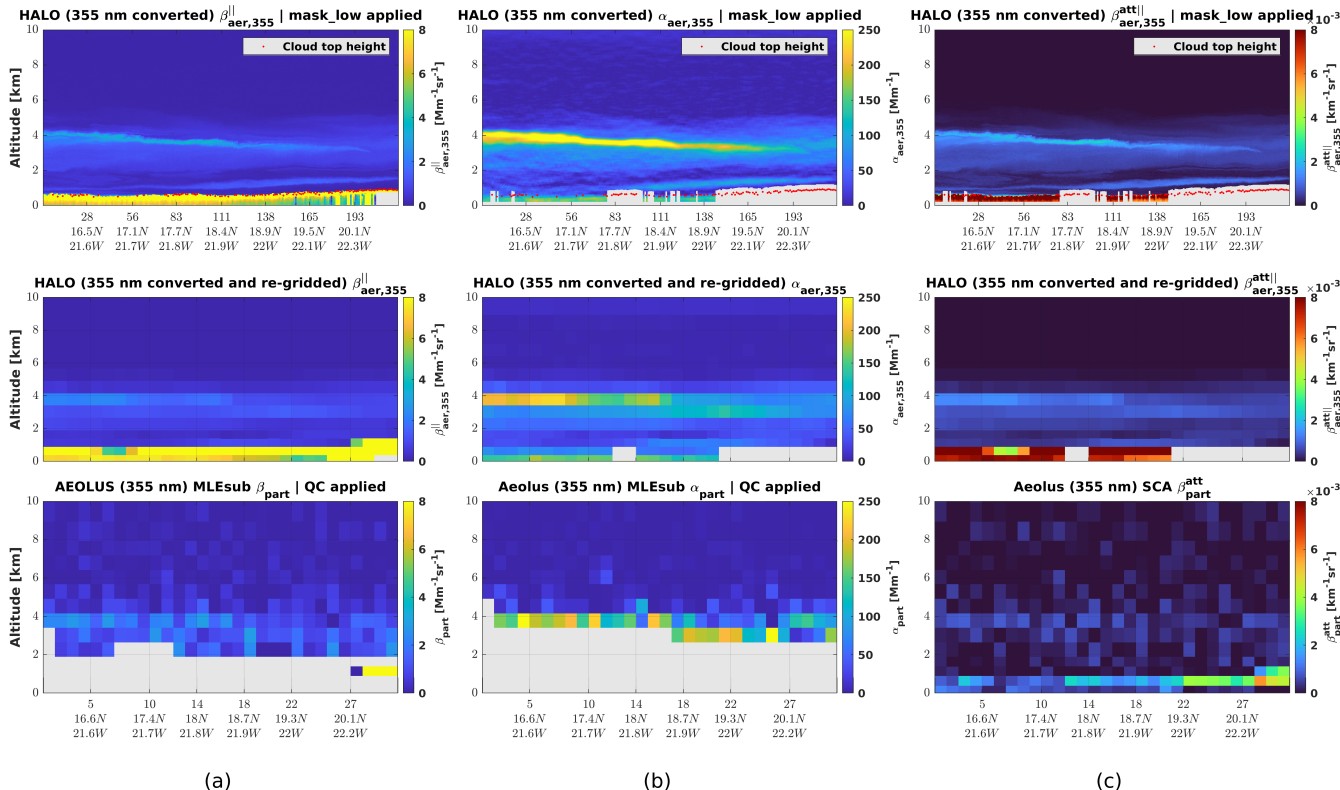

(a)                                    (b)                                    (c)

**Figure 7.** Aeolus MLEsub $\beta_{\text{part}}$ (a), MLEsub $\alpha_{\text{part}}$ (b), and SCA $\beta_{\text{part}}^{\text{att}}$ (c) cross section collocated with HALO 355 nm converted products on 15 September 2022.





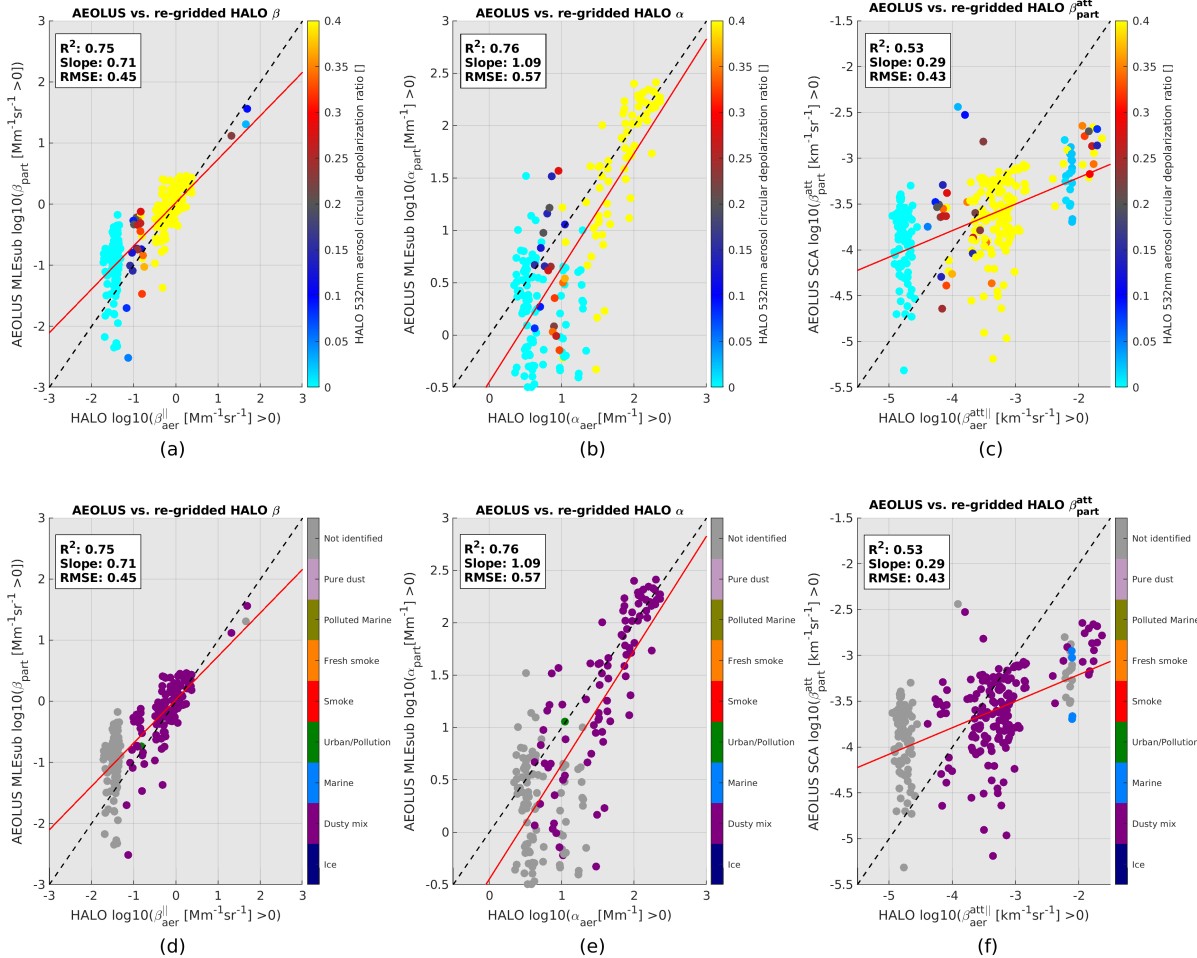

**Figure 8.** Two-dimensional histograms of Aeolus versus HALO 355 nm converted products with HALO $\delta_{\text{circ},532}$ as colors (a-c) and with HALO dominant aerosol type as colors (d-f) on 15 September 2022. A logarithmic transformation has been applied to the signals.



### 5.2.3 Case study 16 September 2022

**Cross section details**. Figures 9a-c show HALO 355 nm converted (top) and re-gridded (middle), and Aeolus 355 nm (bottom) aerosol retrievals for a collocated section of ≈ 538 km horizontal from ≈ [13.2° N - 24.3° W] to ≈ [17.9° N - 25.1° W]. A total of 225 HALO profiles can be re-gridded onto 30 collocated Aeolus measurements. Geolocation offset for the first collocated measurement is equal to ≈ 2 km and initial time offset is equal to ≈ 25 minutes.

**HALO against Aeolus MLEsub**. HALO cloud_top_height superimposed over parallel 355 nm converted particulate backscatter reveals some broken clouds at ≈ 6 km altitude in first tier of the troposheric section. Two main stratified features with strong signatures are particularly noticeable. They are distributed between HALO profiles 29 [13.9° N - 24.3° W] to 142 [16.2° N - 24.8 W] at ≈ 2 km altitude and within last tier of the section by HALO profile ≈ 171 [16.8° N - 24.9° W] at ≈ 3 km altitude. HALO dominant aerosol type mainly classifies the two layers as pure dust (see Appendix A3). Aeolus captures well the two dust layers. They are particularly visible with Aeolus MLEsub extinction coefficient (Fig. 9b bottom, light blue to green color codes). A decrease of HALO 355 nm converted extinction can be observed in top altitude close to profile 86 [15.1° N - 24.6° W].

The scatter diagrams reveal good agreement between HALO and Aeolus MLEsub with $R^2$ up to 0.77 for the backscatter coefficient. The best agreement correspond to non-spherical particles regimes classified as dusty mix by HALO dominant aerosol type product (i.e., dark violet color code in Fig. 10d). The HALO 355 nm converted and Aeolus are in very good agreement for the core dust layer (i.e., yellow point clouds close to the y=x for Figs. 10a-b) classified as pure dust with HALO classification (Fig. 10d-e, rose color code). The HALO aerosol depolarization ratio reaches 0.25 for such layer which is indeed pointing to dust conditions (Freudenthaler et al., 2009; Haarig et al., 2017a, 2022). The ability of Aeolus to retrieve physical retrievals even for highly depolarizing particles such as SAL is then demonstrated. The larger outliers cannot be attributed to an aerosol type, and are distributed in non-aerosol regions with low depolarization ratio for both low and high backscatter (Fig. 10a). This may point to low signal, and bins contaminated by clouds. The scene indeed includes dense water clouds close to ≈ 6 km altitude (Fig. 9a). One could flag the cloudy regions applying a scattering ratio threshold of 5 (Chepfer et al., 2013; Feofilov et al., 2022). If combined with the low scattering removal (i.e. $1.1 < \sigma_{aer,532} <= 5$), this has a positive impact (e.g., increase of $R^2$ from 0.50 to 0.66 and decrease of RMSE from 0.71 to 0.39 for the extinction coefficient).

**HALO against Aeolus SCA**. Marine aerosols are well characterized by Aeolus SCA (blue color code in Fig. 10f), even being located below the pure dust layer which attenuates the signal. Both HALO and SCA reveal broken clouds around ≈ 6 km to ≈ 8 km for the first half of the orbit and the signature of the pure dust layer close to second-last Aeolus measurement 34 is observable within the SCA particulate attenuated backscatter (Fig. 9c, bottom) light blue color code. The scatter diagram shows $R^2$ score of 0.37 and the best agreement is observed for the pure dust and dusty mix regimes (dark violet and rose color codes in Fig. 10f) despite some outliers. The lowest correlation once more corresponds to low signal regions of the atmosphere (gray color code in Appendix A8).





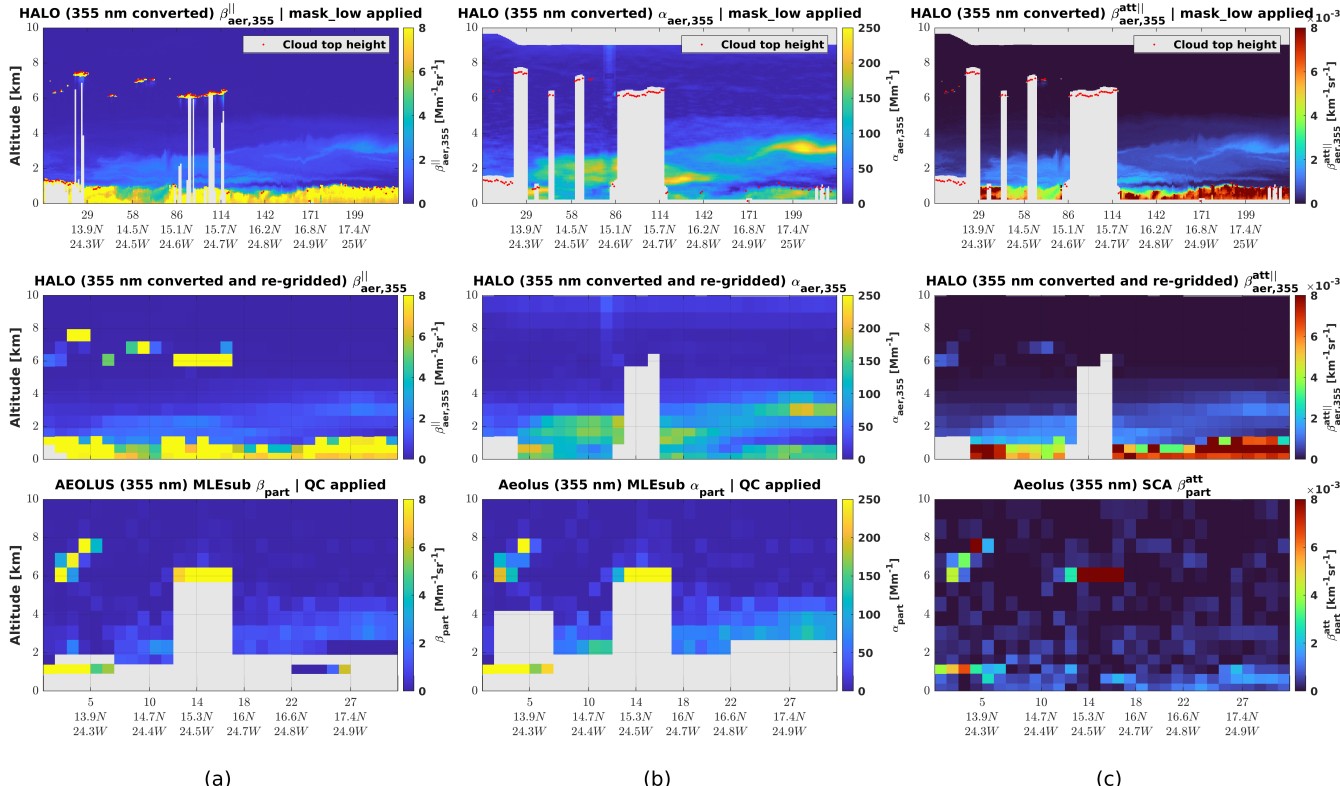

**Figure 9.** Aeolus MLEsub $\beta_{part}$ (a), MLEsub $\alpha_{part}$ (b), and SCA $\beta_{part}^{att}$ (c) cross section collocated with HALO 355 nm converted products on 16 September 2022.





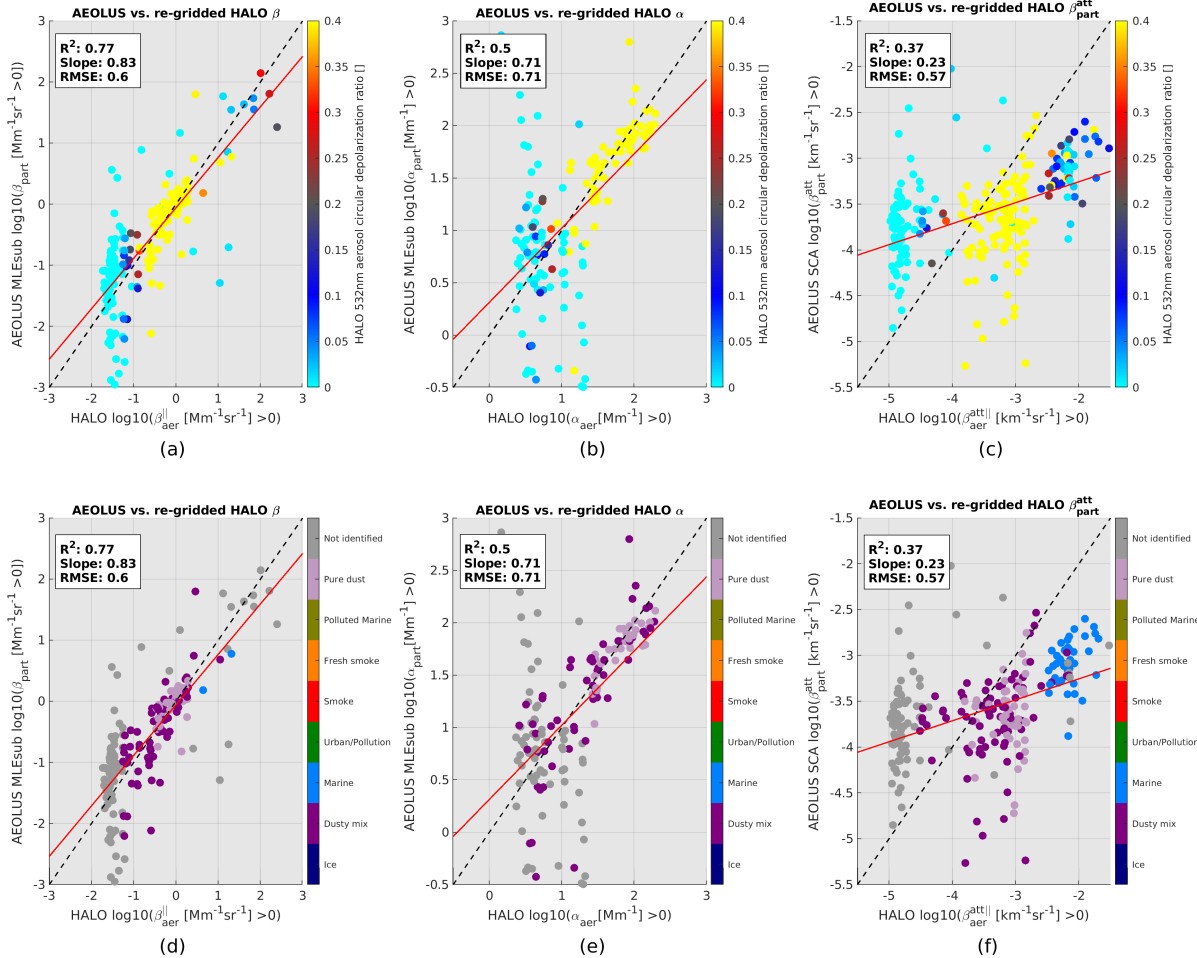

**Figure 10.** Two-dimensional histograms of Aeolus versus HALO 355 nm converted products with HALO $\delta_{\mathrm{circ,532}}$ as colors (a-c) and with HALO dominant aerosol type as colors (d-f) on 16 September 2022. A logarithmic transformation has been applied to the signals.



## 6 Conclusion

The study presents three cross sections of the atmosphere above the Tropical Atlantic as observed by the ALADIN instrument onboard the Aeolus satellite, and reference lidar systems. For the first time, Aeolus aerosol observing capabilities are evaluated

with extended tropospheric sections up to 626 km length with limited geolocation offset (i.e., less than 6 km) and time offset (i.e., less than 30 minutes). The Aeolus L2A aerosol products at 355 nm retrieved with SCA and MLEsub algorithms have been compared to independent measurements from the HALO lidar onboard NASA DC-8 aircraft, the 532 nm signal being converted to 355 nm and re-gridded onto Aeolus sampling. In addition, direct profiles above ground-based Raman lidar Polly$^{XT}$ have been analysed.

Despite different geometry (i.e., nadir-viewing angle for HALO, off-nadir $\approx 35°$ for Aeolus) a solid agreement between the aerosol retrievals is shown for heterogeneous scenes with complex atmospheric conditions. The cross-comparison of the MLEsub backscatter and extinction coefficients for particles with HALO retrievals reveal $R^2$ scores up to 0.83 and 0.76 respectively. When looking at the particulate attenuated backscatter coefficient derived with SCA algorithm, $R^2$ scores reach $\approx 0.69$. A good agreement is observed for highly depolarizing aerosol layers classified as pure dust and dusty mix with the

HALO dominant aerosol type product, and such for each scene on 09 September 2022, 15 September 2022, and 16 September 2022. Higher deviations are observed for aerosol-free regions of the atmosphere where the SNR is expected to be very low. The study reveals that the agreement between the Aeolus and reference measurements can be improved when applying scattering ratio threshold to focus on cloud free regions of the atmosphere with high SNR and higher aerosol loads (e.g., flagging the bins out of a range $1.1 - 5$). This provides further evidence that the SNR and adequate cloud screening approach are important

parameters to consider when using Aeolus L2A product for aerosol profiling.

Optical characteristics of marine aerosols and clouds below dust mixture are well captured by Aeolus (e.g., 09 September 2022) suggesting that the UV laser emissions can penetrate enough aerosol layer distributed in irregular shaped particles with circular aerosol depolarization ratio above 0.30. The opposite condition, i.e., diffuse aerosol layer below broken clouds, should be analysed with caution as the low SNR seems not sufficient to trust retrievals of low backscatter coefficient (e.g., 16

September 2022). Spatial extent of aerosol layers such as SAL are clearly visible with L2A MLEsub as the optimal estimation approach helps to smooth signal over consecutive profiles (i.e., the SCA standard concept corresponding to an along-track averaging of lidar signal resulting in high noise dependency). We then do recommend opting for the MLEsub dataset if focusing on long and homogeneous layers. The SCA algorithm retrievals appears noisier showing higher sensitivity to signal attenuation in low scattering regions of the atmosphere. The SCA also underestimates the particulate attenuated backscatter for both dusty

mix and marine regimes. The Aeolus RBS and their in-orbit deviations must be considered with caution as the better optical properties would be obtained when range bin are not encompassing layer edges or complex mixture. We then do recommend focusing on settings below 1 km thickness when studying vertical sedimentation of aging thin layers. One should note the negative impact of low SNR with thinner bins of 500 m below 2 km altitude. The study shows that the range bin of 750 m correspond to the best agreement with independent lidar measurement, then appearing as the best compromise.



The present study provides further evidence of HSRL benefits for aerosol atmospheric profiling, even for missions initially designed for winds such as Aeolus. The independent measurement of aerosol properties such as particle extinction and backscatter coefficients significantly improve aerosol classification (Baars et al., 2017; Floutsi et al., 2023). The Aeolus MLE-sub extinction is recommended for aerosol layer detection and both the MLEsub backscatter and SCA particulate attenuated backscatter allow to capture water clouds with strong signals below dusty mix. The study illustrates how Aeolus is capable

of measuring optical properties of aerosols, even for highly depolarizing particles once correcting the missing cross-polarized signal. This provides support for further development of new Aeolus product (e.g., co-polarized to total backscatter coefficient using depolarization information from another instrument). The validation of Aeolus L2A SCA and MLEsub products with extended and collocated tropospheric sections with high spectral resolution lidar also supports first estimation of lower limits for the backscatter detection (e.g., $\approx 1.0^{-7}$ m$^{-1}$sr$^{-1}$ for dusty mix focusing on quality flagged MLEsub backscatter

coefficient for particles, and $\approx 2.0^{-4}$ km$^{-1}$sr$^{-1}$ for dust mix focusing on SCA particulate attenuated backscatter). The use of arithmetic mean when re-gridding independent measurement onto Aeolus sampling is straightforward but one could derive the confidence index as function of SNR and aerosol types detected with ad-hoc classification. Vertical sampling is a limitation to bear in mind for new missions such as EarthCARE and Aeolus-2; the finer the vertical resolution with strong signal, the better the measurement will allow analysis of diffuse aerosol plume and mixture where complex atmospheric processes occur. The

importance of a polarization channel for next generation of instrument is also demonstrated as it allows aerosol typing which now becomes a major objective for aerosol missions. Finally, it would be beneficial to reproduce the validation of Aeolus L2A product with independent lidar measurements focusing on the early mission period as a significant signal loss was reported for the ALADIN instrument throughout the satellite lifetime (Lux et al., 2024; Reitebuch et al., 2024). One would expect higher signal and better performance of the L2A algorithms than for the period September 2022 selected for this paper. The public

release of Aeolus reprocessed dataset labelled Baseline 16 and covering 2018 to 2019 is planned for 2025 and can be used for the study.

*Data availability.*  Aeolus Baseline 16 L2A data were obtained from the Aeolus-DISC as part of Cal/Val reprocessing activities which involves TROPOS, DLR, DoRIT, ECMWF, KNMI, CNRS until 31st December 2022, S&T, ABB and Serco. HALO DC-8 data have been downloaded from https://www-air.larc.nasa.gov/cgi-bin/ArcView/cpex.2022#NEHRIR.AMIN/ on May 17, 2023. Data of JATAC 2021 and

2022 can be found at https://earth.esa.int/eogateway/missions/aeolus/data and https://askos.space.noa.gr.

*Author contributions.*  The study was conducted in the frame of Aeolus-DISC as part of L2A processor version 16 development and validation activities. The contributors for L2A SCA and MLE algorithms maintenance and improvement are DT (present), AL (former), TF (former) and AD (former). The operational version of the L2A processor is prepared by DH before deployment at ESA. AN is the principal investigation and technical point of contact for the NASA HALO instrument associated data. This article is part of the special issue «The Joint Tropical

Atlantic Campaign (JATAC)». It is not associated with a conference. The paper was written by DT and reviewed internally by Oliver Reitebuch (German Aerospace Center DLR).



*Competing interests.* The authors declare no competing interests.

*Disclaimer.* The study includes Aeolus aerosol products prepared by the Aeolus DISC (involving ESA, DLR, DoRIT, ECMWF, KNMI, CNRS until 31st December 2022, TROPOS, S&T, ABB and Serco) and the Payload Data Ground Segment (PDGS) through the reprocessing

activities. The corresponding dataset of September 2022 is part of the 4th reprocessing, labelled Baseline 16 and covering July 2019 to April 2023. A first part of this dataset was released to public on May 2024. The Aeolus L2A data are gradually made public since May 2021 through the ESA Aeolus Online Dissemination System ADDF at https://aeolus-ds.eo.esa.int/oads/access/. The Python plotting library Matplotlib (Hunter, 2007) has been used for the study as the Bipolar Colormap which can be downloaded from the MATLAB Central File Exchange at https://www.mathworks.com/matlabcentral/fileexchange/26026-bipolar-colormap.

*Acknowledgements.* We thank all previous contributors to the L2A product development and validation from Meteo France CNRM/CNRS (Pierre Flamant, Marie-Laure Denneulin, Mathieu Olivier, Vincent Lever, Pauline Martinet, Hugo Stieglitz, Ibrahim Seck). We acknowledge TROPOS members involved in the Ocean Science Centre Mindelo (OSCM) instrumentation maintenance Ronny Engelmann, Dietrich Althausen and Annet Skupin. We also aknowledge the HALO team for the collection of CPEX-CV dataset as well as the entire CPEX-CV science, instrument, and aircraft operations teams for the successful execution of the CPEX-CV campaign. We also thank Aaron Piña and

Edward P. Nowottnick from NASA for their support within the JATAC Campaign.



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



## Appendix A

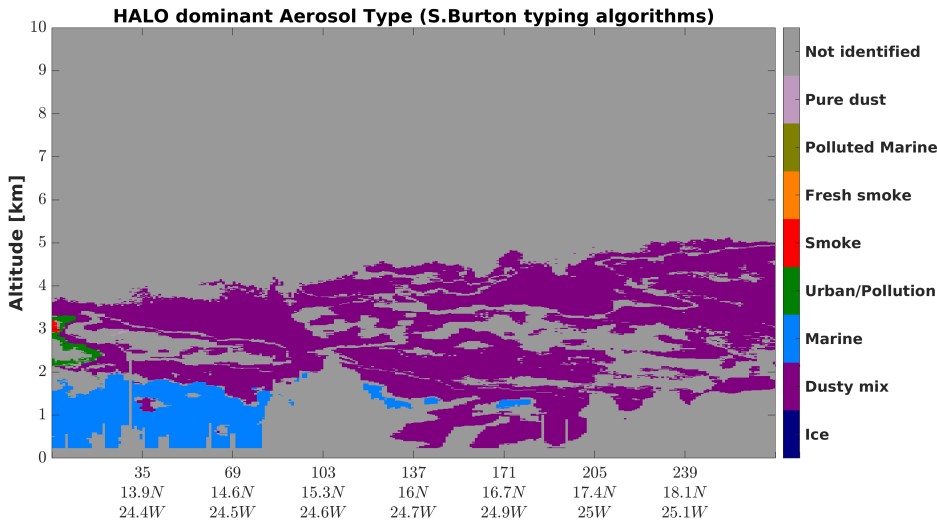

**Figure A1.** HALO 532 nm dominant aerosol type for 09 September 2022, aerosol-free regions as Not identified.



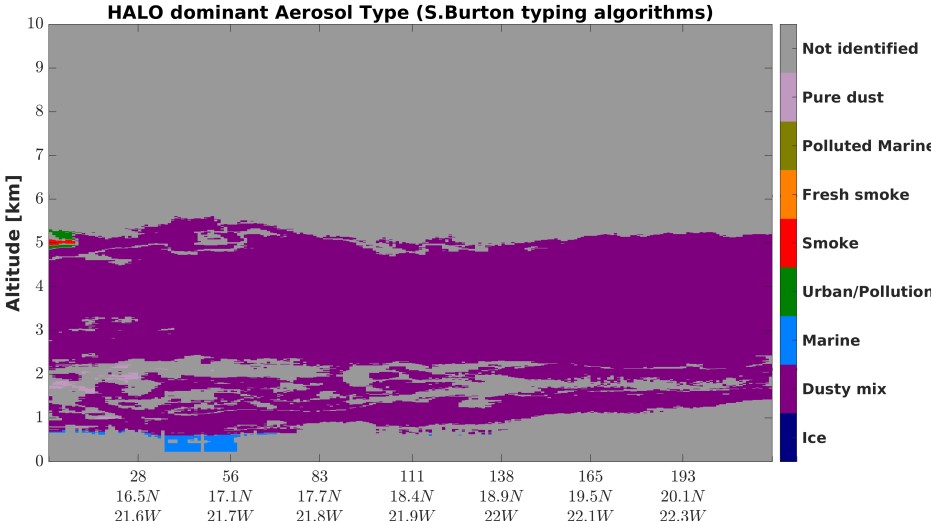

**Figure A2.** HALO 532 nm dominant aerosol type for 15 September 2022, aerosol-free regions as Not identified.



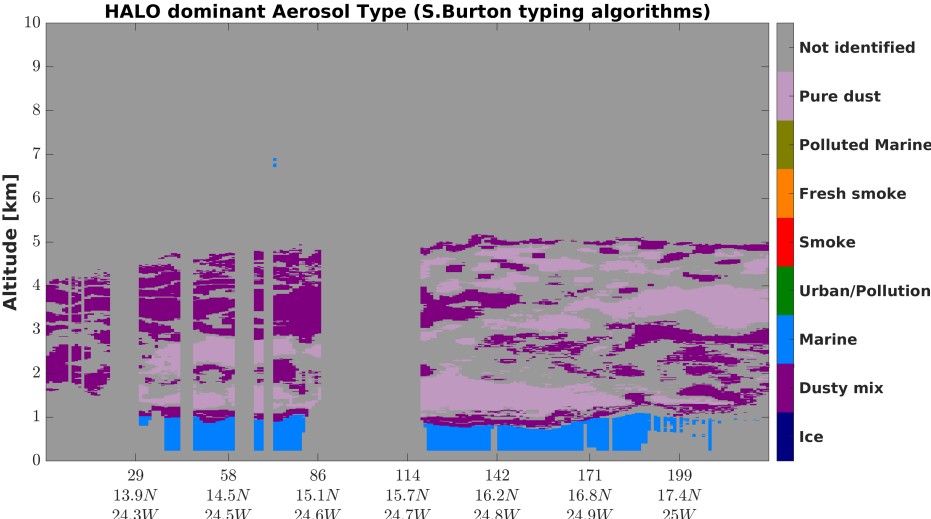

**Figure A3.** HALO 532 nm dominant aerosol type for 16 September 2022, aerosol-free regions as Not identified.



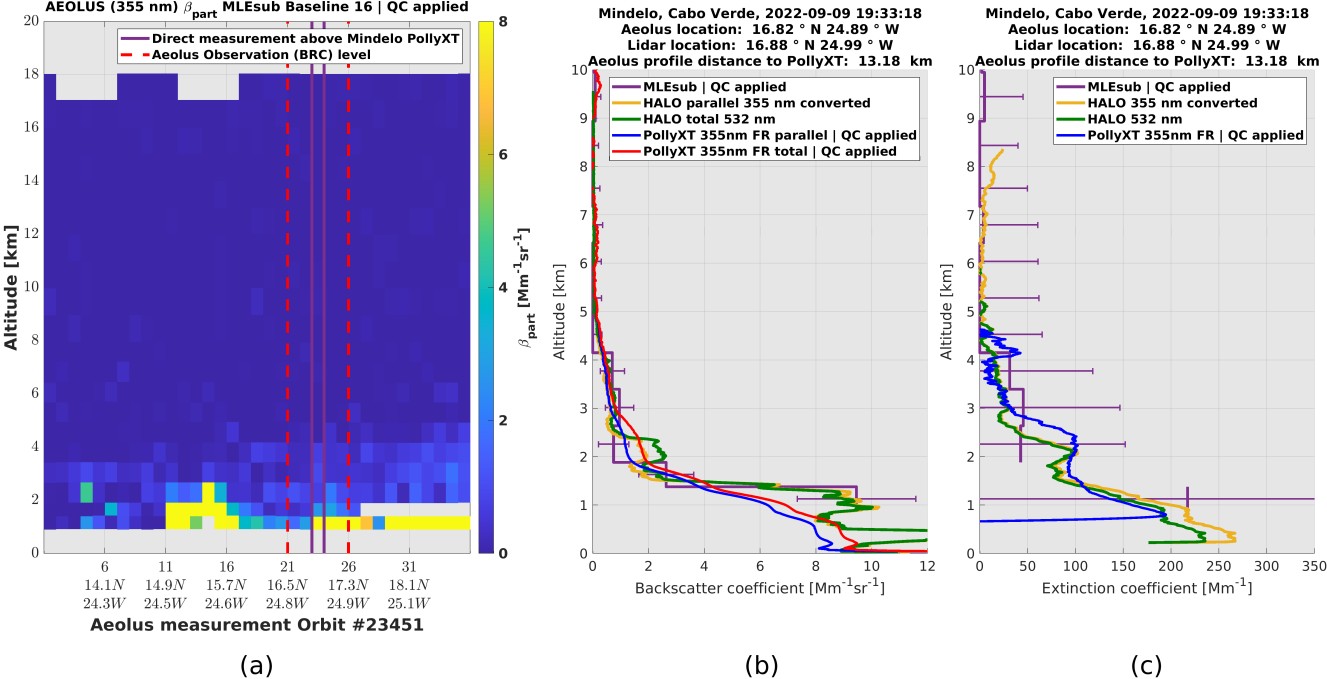

**Figure A4.** Aeolus MLEsub $\beta_{\text{part}}$ on 09 September 2022 (a) and cross of 2D profiles above Mindelo, Cabo Verde with HALO and Polly$^{\text{XT}}$ for MLEsub $\beta_{\text{part}}$ (b) and MLEsub $\alpha_{\text{part}}$ (c).



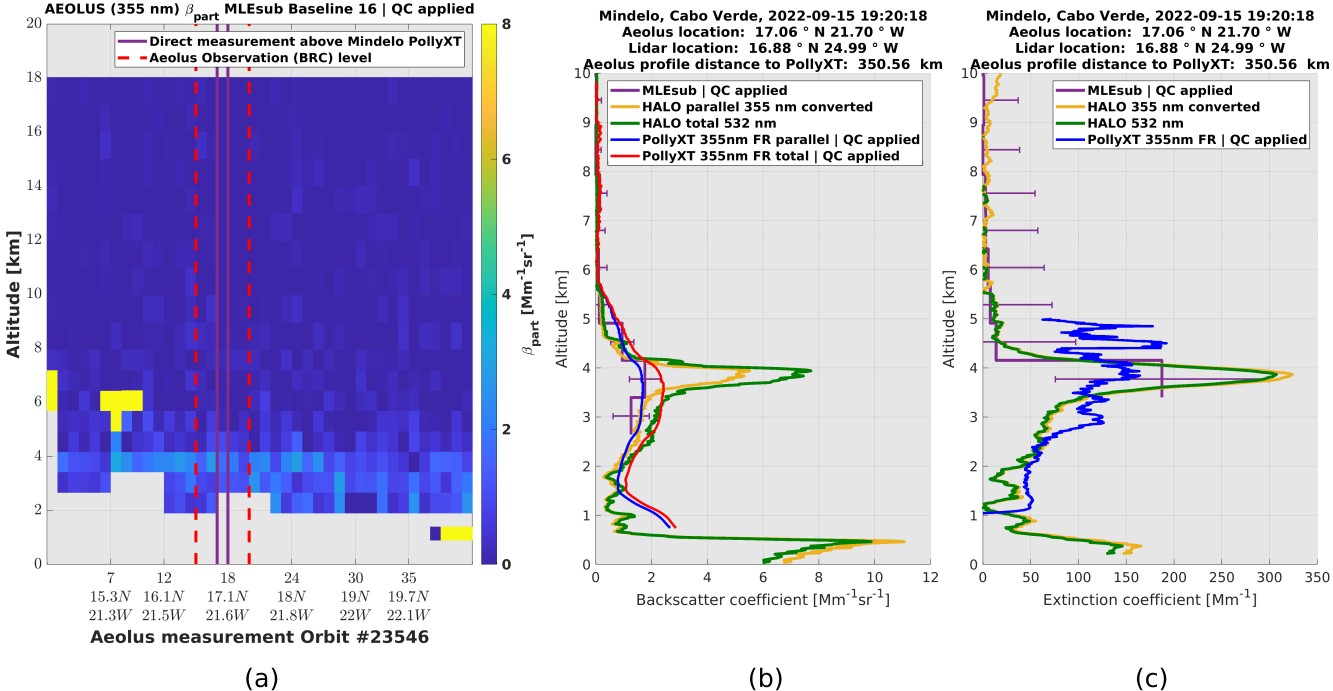

**Figure A5.** Aeolus MLEsub $\beta_{\text{part}}$ on 15 September 2022 (a) and cross of 2D profiles above Mindelo, Cabo Verde with HALO and Polly$^{\text{XT}}$ for MLEsub $\beta_{\text{part}}$ (b) and MLEsub $\alpha_{\text{part}}$ (c).





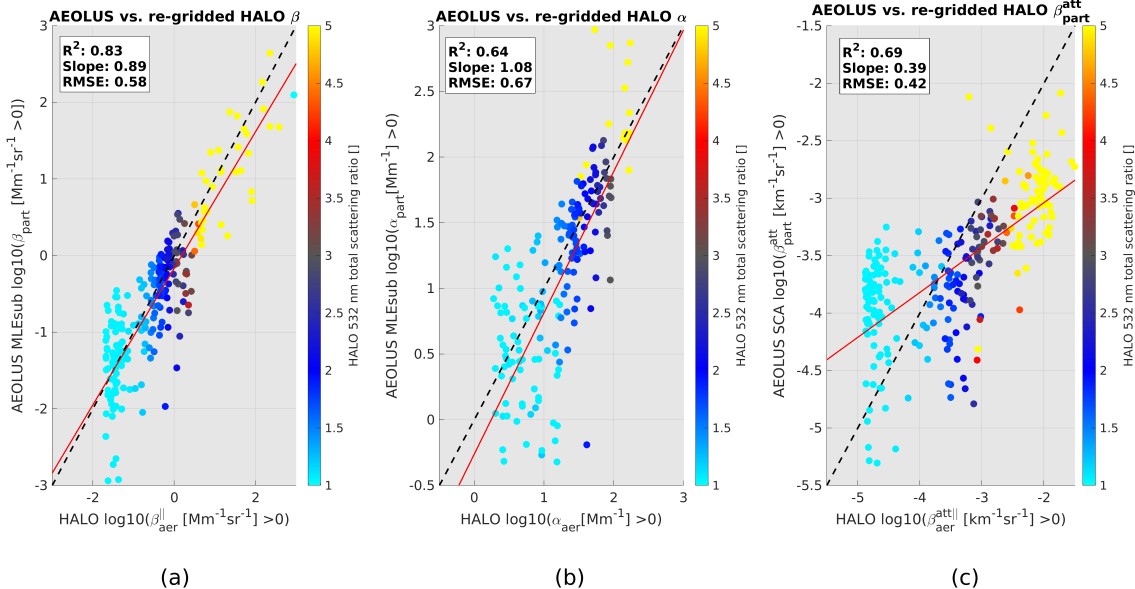

**Figure A6.** Two-dimensional histograms of Aeolus MLEsub $\beta_{part}$ (a), MLEsub $\alpha_{part}$ (b), and SCA $\beta_{part}^{att}$ (c) versus HALO 355 nm converted products with HALO $\sigma_{aer,532}$ as colors on 09 September 2022.





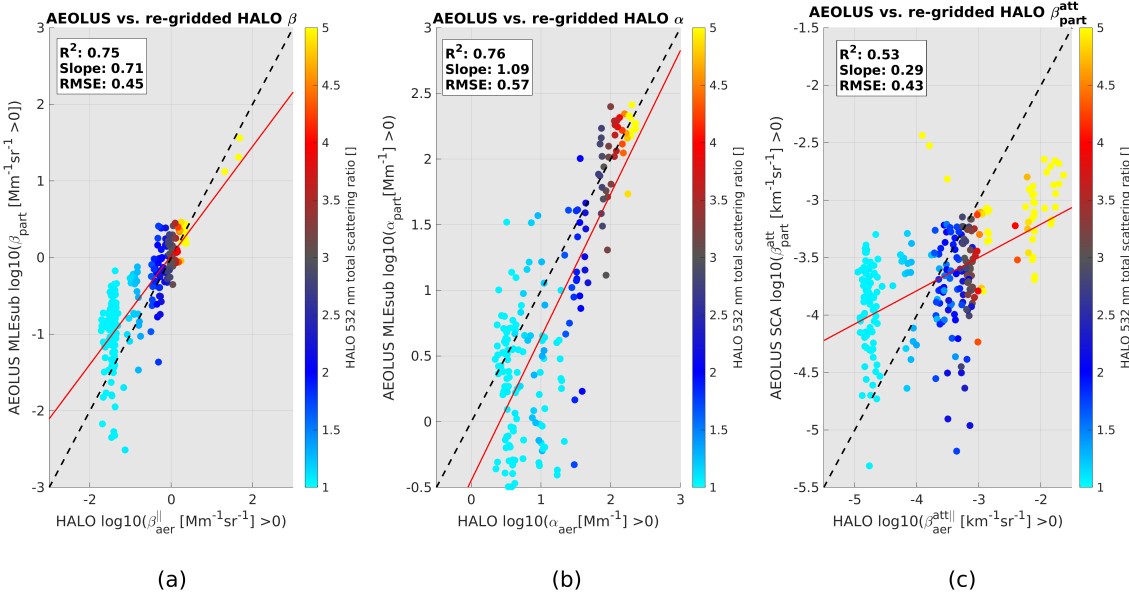

**Figure A7.** Two-dimensional histograms of Aeolus MLEsub $\beta_{part}$ (a), MLEsub $\alpha_{part}$ (b), and SCA $\beta_{part}^{att}$ (c) versus HALO 355 nm converted products with HALO $\sigma_{aer,532}$ as colors on 15 September 2022.





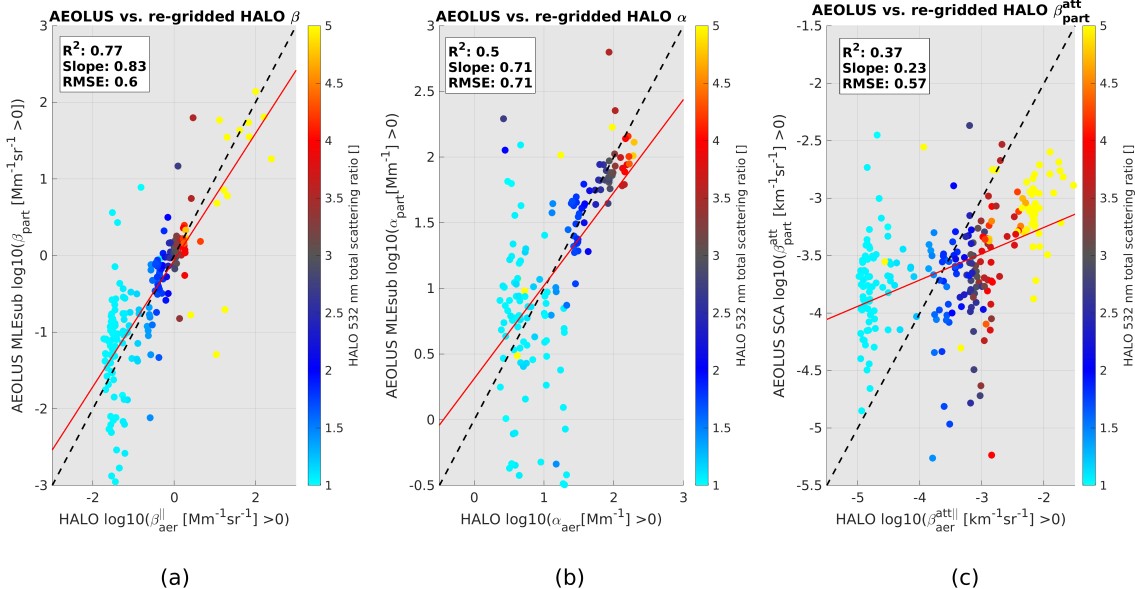

(a)  (b)  (c)

**Figure A8.** Two-dimensional histograms of Aeolus MLEsub $\beta_{\text{part}}$ (a), MLEsub $\alpha_{\text{part}}$ (b), and SCA $\beta_{\text{part}}^{\text{att}}$ (c) versus HALO 355 nm converted products with HALO $\sigma_{\text{aer,532}}$ as colors on 16 September 2022.