# Peer review of "Cross validations of the Aeolus aerosol products and new developments with airborne high spectral resolution lidar measurements above the Tropical Atlantic during JATAC"

_EGUsphere, 2025_

## Author Response (AR1)

Atmos. Meas. Tech. Discuss., response file to the referees
https://doi.org/10.5194/egusphere-2025-462 © Author(s) 2025. This work is distributed under the Creative Commons Attribution 4.0 License.

**Response file to the referees on "Cross validations of the Aeolus aerosol products and new developments with airborne high spectral resolution lidar measurements above the Tropical Atlantic during JATAC " by Dimitri Trapon et al., Atmos. Meas. Tech. Discuss., https://doi.org/10.5194/egusphere-2025-462, 2025**

**Response to the referees :**

Dear editorial board and referees,

The present document includes the replies to RC1 and RC2 after a brief description of the modifications made within the revised manuscript, the sort summary and the abstract. All the text corrections (i.e. , including deleted and added text from pre-print manuscript) are indicated in blue in the attached pdf document
« Cross_validations_of_the_Aeolus_aerosol_products__Revised_Track_Changes.pdf ».

- The following corrections have been made following the recommendation by the editorial board with pre-print manuscript acceptance : credits added for Fig. 2, revision of Figs. 1, 4, 5, 7, 9, A4, A5 with selection of Colorblind safe color schemes (i.e., avoiding as much as possible the overimposed red and green color codes, and restricting the MATLAB colormaps to parula, turbo, and bipolar), correction of typo for the Short Summary below (i.e., correction in bold).
- It has to be noted that the authors propose the inclusion of a new co-author whose contribution allowed to positively reply to one of the referees' comment and suggestion.
- Addition of the references below :
  - Barton-Grimley, R. A., Nehrir, A. R., Kooi, S. A., Collins, J. E., Harper, D. B., Notari, A., Lee, J., DiGangi, J. P., Choi, Y., and Davis, K. J.: Evaluation of the High Altitude Lidar Observatory (HALO) methane retrievals during the summer 2019 ACT-America campaign, Atmospheric Measurement Techniques, 15, 4623–4650, https://doi.org/10.5194/amt-15-4623-2022, 2022.
  - Bruneau, D., Pelon, J., Blouzon, F., Spatazza, J., Genau, P., Buchholtz, G., Amarouche, N., Abchiche, A., and Aouji, O.: 355-nm high spectral resolution airborne lidar LNG: system description and first results, Appl. Opt., 54, 8776–8785, https://doi.org/https://doi.org/10.1364/AO.54.008776, 2015.
  - Fehr, T., McCarthy, W., Amiridis, V., Baars, H., von Bismarck, J., Borne, M., Chen, S., Flamant, C., Marenco, F., Knipperz, P., Koopman, R., Lemmerz, C. L., Marinou, E., Močnik, G., Parrinello, T., Piña, A., Reitebuch, O., Skofronick-Jackson, G., Zawislak, J., and Zenk, C.: The Joint Aeolus Tropical Atlantic Campaign 2021/2022 Overview– Atmospheric Science and Satellite Validation in the Tropics, EGU General Assembly 2023, https://doi.org/https://doi.org/10.5194/egusphere-egu23-7249, 2023.
  - Gebauer, H., Floutsi, A. A., Haarig, M., Radenz, M., Engelmann, R., Althausen, D., Skupin, A., Ansmann, A., Zenk, C., and Baars, H.: Tropospheric sulfate from Cumbre Vieja (La Palma) observed over Cabo Verde contrasted with background conditions: a lidar case study of aerosol extinction, backscatter, depolarization and

lidar ratio profiles at 355, 532 and 1064 nm, Atmospheric Chemistry and Physics, 24, 5047–5067, https://doi.org/10.5194/acp-24-5047-2024, 2024.

- o Turk, F. J., Hristova-Veleva, S., Durden, S. L., Tanelli, S., Sy, O., Emmitt, G. D., Greco, S., and Zhang, S. Q.: Joint analysis of convective structure from the APR-2 precipitation radar and the DAWN Doppler wind lidar during the 2017 Convective Processes Experiment (CPEX), Atmospheric Measurement Techniques, 13, 4521–4537, https://doi.org/10.5194/amt-13-4521-2020, 2020.
- o Winker, D. M., Vaughan, M. A., Omar, A., Hu, Y., Powell, K. A., Liu, Z., Hunt, W. H., and Young, S. A.: Overview of the CALIPSO Mission and CALIOP Data Processing Algorithms, Journal of Atmospheric and Oceanic Technology, 26, 2310 – 2323, https://doi.org/10.1175/2009JTECHA1281.1, 2009.

- In addition to the replies to RC1 and RC2, some minor re-phrasing corrections or additions have been made : lines 22, 37, 41, 56, 84, 106, 117, 118, 187, 280, 290, 291, 292, 307, 308, 316, 326, 327, 384, 390, 411, 412, 413, 44, 447, 493, 564.

**Short Summary :**

The study highlights how aerosol measurements from aircraft can be used in synergy with ground-based observations to validate the **European Space Agency**'s Aeolus satellite aerosol product above the Tropical Atlantic. For the first time, collocated sections of the troposphere up to 626 km long are crossed. Combining measurements from satellite, aircraft, and ground-based instruments allows to characterize the optical properties of the observed dust particles emitted from the Sahara Desert.

**Abstract :**

The Joint Aeolus Tropical Atlantic Campaign (JATAC) conducted 2022 in Cabo Verde has provided quantitative lidar measurements, in particular from the NASA Langley High Altitude Lidar Observatory (HALO) on-board DC-8 aircraft, for process level understanding of tropical dynamics, as well as for satellite validation. For the first time, optical properties of particles (i.e., backscatter, extinction, attenuated backscatter **coefficients**, and depolarization **ratios**) have been measured for extended tropospheric sections collocated with the Aeolus satellite overpasses with limited geolocation and time offsets. This has contributed to the evaluation of the Aeolus Level-2A (L2A) aerosol optical properties product. In addition, localized aerosol profiles were measured by the ground-based multiwavelength Raman polarization and water-vapor lidar Polly$^{XT}$.

With this study, we assess the quality of the Aeolus L2A product retrieved with the Standard Correct Algorithm (SCA) and the Maximum Likelihood Estimation (MLE) as part of the September 2022 dataset reprocessed with the L2A processor version 16. The focus is given to the 355 nm aerosol retrievals given at finer horizontal resolution, i.e., so-called Aeolus measurement level of $\approx$ 18 km. They are compared to the 532 nm HALO airborne profiles **that** are converted to 355 nm using the backscatter Ångström exponent. HALO and Polly$^{XT}$ polarization lidars also provide insights about the L2A algorithms limitations when looking at non-spherical particles such as Saharan dust. Even though having no cross-polarized component the Aeolus measurements can be corrected using collocated observations with such instruments that include both co-polarized and cross-polarized components of the backscattered light. Moreover the cross **validation** with independent lidar measurements **enables** to estimate lower limits for Aeolus backscatter detection.

Atmos. Meas. Tech. Discuss., referee comment RC1
https://doi.org/10.5194/egusphere-2025-462 © Author(s) 2025. This work is distributed under the Creative Commons Attribution 4.0 License.

Comment on amt-2025-462

Anonymous Referee #1

Referee comment on "Cross validations of the Aeolus aerosol products and new developments with airborne high spectral resolution lidar measurements above the Tropical Atlantic during JATAC " by Dimitri Trapon et al., Atmos. Meas. Tech. Discuss., https://doi.org/10.5194/egusphere-2025-462-RC1, 2025

**A Review of "Cross validations of the Aeolus aerosol products and new developments with airborne high spectral resolution lidar measurements above the Tropical Atlantic during JATAC" by Dimitri Trapon et al.**

This paper provides the most robust validation efforts of the Aeolus aerosol data products to date. The paper is well written, clear, and provides results that are important for the interpretation of Aeolus aerosol data products. It deserves to be published after a few minor revisions that I believe will strengthen the paper.

My 4 main comments are:

1) Importance of Aeolus aerosol products: The lidar and wind communities view Aeolus as a wind lidar first and foremost (as they should). The aerosol community typically uses CALIOP for aerosol lidar data. Why should either community use the Aeolus aerosol data products? Some in the aerosol community may be hesitant to use Aeolus products due to the coarse resolutions and low sensitivity to faint aerosols. However, there are good reasons to use these products. As one example, simultaneous aerosol and wind lidar profiles can inform aerosol transport/advection. This is especially true for dust in the SAL. I recommend the authors address this early in the Introduction with a dedicated paragraph.

The value of Aeolus aerosol profiling comes with the ability to separate the contributions from molecules and particles contrary to elastic backscatter lidars such as CALIOP/CALIPSO. Aeolus therefore measures both the particles backscatter and extinction coefficients independently, leading to no ambiguities in aerosol and clouds optical properties, and without using a-priori hypothesis on extinction-to-backscatter ratio. We emphasised this point in the introduction, and we added a sentence about the Aeolus capacity to provide simultaneous aerosol and wind lidar profiles.

2) Converting HALO data to 355 nm: This is probably my biggest concern reading the paper, given it is so important to the entire results section. I understand why the authors used the median backscatter and extinction Angstrom exponents from PollyXT as they are the most applicable. However, the statistics reported in Section 4.1 have a very wide range. This made me wonder several things: (1) what do the histograms of these Angstrom exponents look like? (2)

how sensitive is the 355 nm conversion to these Angstrom exponents? (3) Would there be less range/variability if the authors only used the PollyXT data from those specific dates to convert the HALO from 532 to 355 nm? Then when I look at Figures 4a and 4b, I see that there is much better agreement between the PollyXT and MLEsub backscatter and extinction profiles than the HALO 355 nm and MLEsub, especially between 2.5 and 3.5 km in the backscatter. I highly recommend the following: (1) Instead of using the entire month of September for the PollyXT Angstrom exponent calculations, try using just that specific day or +/- 1-2 days around that day (if you need more data). See if that reduces the range/variability of the values. (2) Provide the histograms of the Angstrom exponents for whatever set of PollyXT data you use for the conversion in an Appendix. (3) Add error bars to the yellow HALO 355 nm profiles in Figures 4a/b (and Figures A4 and A5), similar to what you have for the purple MLEsub profiles, that show use the 25th and 75th percentiles. (4) Add some discussion in Section 5.2 about what these uncertainties in the HALO 355 nm conversion mean for the cross-section comparisons. Note: the MLEsub and PollyXT 355 nm data show very good agreement (Fig 4, A4, A5), especially for the backscatter. This is a should give users confidence in using the Aeolus MLE aerosol data!

The Angstrom exponents' median values of the monthly profiles were initially selected as the number of valid profiles per day vary, and because it can be considered well representative for the conditions observed for September 2022, i.e., with occurrence of Saharan Air Layer (SAL) in the troposphere and the close-to-zero reported values being in accordance with the literature for mineral dust.

As recommended, the Angstrom exponents' median values of the daily profiles for 09, 15 and 16 September 2022 have been derived from the PollyXT measurements and used to reprocess the 532 nm to 355 nm conversion of the HALO atmospheric products. A new co-author who derived the Angstrom exponents values has been added (i.e., Moritz Haarig from TROPOS). This results in a much better agreement between the MLEsub and HALO parallel 355 nm backscatter profiles in Figs. 4b below, A4b and A5b; the HALO and PollyXT parallel 355 nm backscatter profiles being now really close (i.e., offset less than $\approx 0.1$ Mm$^{-1}$sr$^{-1}$ for high aerosol loads and inner SAL around 3 km altitude on 16 September 2022). One should also note a better agreement in the Marine Boundary Layer (MBL). It has to be noted that the use of Angstrom exponents daily median instead of monthly median results in a minor impact onto the metrics for the inter-comparison between Aeolus and HALO converted and re-gridded products (e.g., decrease of 0.01 for RMSE with MLEsub backscatter coefficient on 09 September 2022).

[Figure]

Figure 4b. Parallel backscatter coefficient for particles at 355 nm derived with Aeolus MLEsub algorithm (i.e., violet), PollyXT (i.e., blue) and HALO (i.e., dark yellow) on 16 September 2022.

The section 4.1 and the Figs. 4b-c, A4b-c, A5b-c have been revised accordingly. The Table 1 as the Figs. 5, 6, 7, 8, 9, 10, A4, A5, A6, A7, A8 have also been updated.
The errors bars have been added to the yellow HALO 355 profiles in Figs. 4b-c, A4b-c and A5b-c. A statement introducing the uncertainties induced by the wavelength conversion for HALO and the deviations per Aeolus range bin via the re-gridding has been added in section 5.1.

It has to be mentioned that the statistical metrics presented in the pre-print correspond to the Pearson Correlation Coefficient (PCC) and the unsystematic Root Mean Square Error (RMSEu) instead of $R^2$ and RMSE. Nevertheless, as we think that the use of $R^2$ and RMSE is preferable (i.e., the focus here being more the assessment of the deviations between HALO and Aeolus using RMSE than the characterization of the dispersion from the regression line with RMSEu, and the $R^2$ being often used as complement) the metrics have been corrected. This does not invalidate the observations and conclusions.

3) Cloud contamination in aerosol retrievals: On lines 248-249, you say "Invalid HALO measurements that correspond to localized conditions, e.g., below dense clouds or within the PBL, and reported as NaN values in the ready-to-use products are not taken into account." Does this just mean those bins aren't included in the re-gridding? What about scenes with thin cumulus clouds? How are conditions handled when half of the HALO obs in an Aeolus bin are NaN or in thin/small cloud and half are not (in a dust plume)? For example, how are the clouds (red dots) on the left side of the high-resolution HALO backscatter curtain (Figure 5, top left)

handled when re-gridding? Please add some text that describes how these situations are handed and describe the consequences of including high resolution cloud bins in coarse resolution re-gridded data.

The HALO bins with NaN values are not included in the re-gridding, hence a gap filling below dense clouds (e.g., Fig. 9a). One could expect a less representative cross-comparison between HALO and Aeolus for such regions if compared to fully valid bin. This statement has been added to the paragraph in section 4.2. The re-gridding with scattered or broken clouds in regions highlighted with HALO cloud_top_height product in Figs. 5, 7 and 9 results in averaging dilution for cloud echos as the Aeolus range bin may encompass surrounding lower backscatter signal (e.g., Fig. 3b at altitude 6 km and HALO profiles 86 to 114). The averaging dilution is expected to be more pronounced with thicker range bin and with coarser resolution at observation level (i.e., Basic Repeat Cycle BRC). The authors then do recommend to use the Aeolus measurement level (i.e., sub-BRC) products when studying clouds. The point has been stressed into section 5.2.1 and within the conclusion.

4) Conclusion & relating to Aeolus 2: There are a few things that could be added or reworded to really help Aeolus aerosol data users and drive home the needed capabilities of the Aeolus-2 follow-on.

a. Lines 396-400: The sentence that starts "The study reveals that the agreement between the Aeolus and reference measurements can be improved when applying scatter ratio threshold…" This is a key finding of the study. Can you be more explicit to Aeolus data users as to how to filter the data to get the highest quality (only use backscatter values above X.X and scattering ratios below X.X) ?

The highest quality of Aeolus L2A data is obtained when applying the pre-defined Quality Check (QC) flags based on Signal-to-Noise Ratio (SNR) and error estimates as described in the L2A User Guide v2.2. When focusing on high aerosol load only, the use of a scattering ratio threshold may help to discard the background noise and the cloud contribution. The corresponding sentence has been modified.
The recommendation about the lower limit for the backscatter detection is given in the last paragraph of the conclusion.

b. Lines 415-417: The sentence that starts with "The present study provides further evidence of HSRL benefits for aerosol atmospheric profiling…" should be rewritten. It comes across to the reader as if Aeolus demonstrates HSRL capabilities for aerosol typing but given the results of this study and the lack of polarization, I don't think that is what the authors are trying to say. I suggest changing this sentence to something like "The present study demonstrates the independent HSRL measurement of aerosol properties such as particle extinction and backscatter coefficients, even for missions initially designed for winds such as Aeolus. The collocation of the Aeolus data with the HALO polarization measurements demonstrates the aerosol typing capabilities and provides further evidence of the need for Aeolus-2 to have polarization capabilities."

CORRECTED

c. Lines 427-429: The sentence "Vertical sampling is a limitation to bear in mind for new missions such as EarthCARE and Aeolus-2…". I think this is another key takeaway from this paper and something that should be considered for Aeolus-2. The comments raised in #3 (cloud

contamination in aerosol retrievals) are a great example of the importance of the higher resolutions.

This takeaway message has been strengthened in parallel to the additions for comment 3).

d. I think this paper can provide great examples of why polarization capabilities and finer resolutions are needed for Aeolus 2. I highly recommend a sentence at the beginning of last paragraph of the conclusion that explicitly states

This takeaway message has been strengthened in parallel to the additions for comments 3) and 4)c.

Other minor issues to be addressed are:

1) That vs which: The word "which" should be changed to "that" in lines 11, 367, 376

CORRECTED

2) Allows to: There are several places where the authors use the phrase "allows to", which reads awkwardly to me. I suggest change this to "enables" or "allows for estimates of". This occurs in lines 15, 44, 53, 103, 253, 264, 430

CORRECTED

3) Line 35: MLE acronym is used but not defined in the line above.

CORRECTED

4) Lines 59. Delete "It is also important to point that" as it is unnecessary.

CORRECTED

5) Line 111: It says "Baseline 16 does not include QC flags for SCA particulate attenuated backscatter which then contains non-physical values (i.e., negative) because of non-perfect cross-talk correction. It was decided to flag the negative values for the study". I recommend adding some discussion as to what it means to flag this data and instructions for potential data users. It would be even better if you could provide a figure in an Appendix that demonstrates the importance of flagging these negative values.

Negative values for SCA particulate attenuated backscatter (up to ~ 40 %) are distributed in low SNR regions with aerosol-free conditions. The information has been added to the paragraph in section 2.1.

6) Line 120: HALO has already been spelled out and the acronym defined, so I would use the acronym here. The same thing for line 139.

CORRECTED

7) Figure 2: The text in the key isn't readable (too small). Are the Aeolus overpasses/tracks exactly the same every Thurs, Fri, and Sat? If not, please provide the exact dates of interest. If no Saturday overpasses are used in the paper, I recommend removing it from the figure.

The Aeolus overpasses selected for the study only correspond to Thursday (i.e., 15 September 2022) and Friday (i.e., 09 and 16 September 2022). The Saturday overpasses (i.e., Fig. 4c) have then been removed from the figure 4.

8) Line 128: CPEX-CV has already been spelled out and the acronym defined, so I would use the acronym here.
CORRECTED

9) Line 140: There should be a comma before which. Same thing for line 145, 178, 296, 307, 332, 430
CORRECTED

10) Line 181: The phrase "It is proposed" makes it sound like you are asking to do the conversion this way. I suggest changing this phrase to "It is necessary"
CORRECTED

11) Line 217: The authors say "This constant is added to the $\alpha_{mol,355}$ in Eq. (11)". Do they mean that the constant has been added to the equation or do they mean that value was used for computing $\alpha_{mol,355}$? The way this is stated is confusing to me.
The constant has been added to the $\alpha_{mol,355}$ derived in Eq. (11) when calculating the HALO 355 nm attenuated backscatter coefficient for particles in Eq. (12). The sentence has been corrected.

12) Line 232: The phrase "from an over-sampled $\approx$ 15 m resolution" refers to the vertical resolution, correct? If so, please add the word "vertical" before resolution.
Yes, CORRECTED

13) Line 259: The authors say "Figure 4a illustrates how the Aeolus MLEsub $\beta_{part}$ up to 18 km altitude helps assessing the particle-free conditions above the DC-8 flying at $\approx$ 10.7 km altitude for the 16 September 2022 case." I don't understand how Figure 4a illustrates this? I don't see any backscatter signal that looks like a layer above 10 km. Maybe you mean to say that the Aeolus MLEsub $\beta_{part}$ up to 18 km altitude confirms the assumption of particle-free conditions above the DC-8?
The Aeolus MLEsub $\beta_{part}$ up to 18 km altitude in Figs. 4a, A4a, A5a confirm the particle-free conditions above the DC-8 as no backscatter signal corresponding to aerosol layers or clouds can be observed above 10 km altitude (i.e., with close-to-zero values in dark blue color code). The sentence has been corrected.

14) Line 266: the sentence "The HALO parallel 355 nm profile agrees with Aeolus MLEsub (i.e., violet)." Is too vague. The 2 profiles agree above 4 km, but below 4 km the HALO backscatter and extinction is 25-50% higher by my eye.
CORRECTED

15) Line 281: The authors use the phrase "look solid". I suggest a more formal phrase, such as "are robust". The word solid is used also in line 337 and 390
CORRECTED

16) Lines 390-391: I don't consider these scenes to be heterogeneous, especially considering profiles with clouds have been screened from the statistics analysis shown in Figures 5-10. I recommend changing the phrase "a solid agreement between the aerosol retrievals is shown for heterogeneous scenes with complex atmospheric conditions" to something like "good agreement

between the aerosol retrievals is shown for homogeneous, optically thick aerosol layers."
CORRECTED

Atmos. Meas. Tech. Discuss., referee comment RC2
https://doi.org/10.5194/egusphere-2025-462 © Author(s) 2025. This work is distributed under
the Creative Commons Attribution 4.0 License.

Comment on amt-2025-462

Anonymous Referee #2

Referee comment on "Cross validations of the Aeolus aerosol products and new developments
with airborne high spectral resolution lidar measurements above the Tropical Atlantic during
JATAC " by Dimitri Trapon et al., Atmos. Meas. Tech. Discuss.,
https://doi.org/10.5194/egusphere-2025-462-RC2, 2025

**Review on the manuscript entitled "Cross validations of the Aeolus aerosol products and new developments with airborne high spectral resolution lidar measurements above the Tropical Atlantic during JATAC" by Trapon D. et al.**

The manuscript presents the results of a validation study on the Aeolus Level 2A products using as reference collocated airborne lidar measurements from the NASA's HALO instrument onboard the DC-8 aircraft that was deployed at the CPEX-CV campaign (Cabo Verde, summer 2022) in the framework of the Joint Aeolus Tropical Atlantic Campaign (JATAC). The study is focused on evaluating the performance of the Aeolus L2A products from the reprocessed Aeolus dataset with the latest available version (Baseline 16) that are provided in the highest available horizontal resolution (~18 km) from the Standard Correct Algorithm (SCA) and the Maximum Likelihood Estimation (MLE) algorithms. Moreover, ground-based lidar observations from the PollyXT lidar during the JATAC campaign have been used to facilitate the wavelength and the total to parallel conversions for the harmonization of the HALO products with Aeolus. The manuscript demonstrates the capability of Aeolus to retrieve profiles of the aerosol properties (L2A products) in higher horizontal resolution and provides valuable recommendations for the Aeolus data users.

Overall, the manuscript is well structured and well written even though I have the feeling that some parts could be further explained and/or discussed. The scientific significance makes the manuscript suitable for publication in AMT, after some minor revisions have been considered from the authors.

**Specific comments**

Line 15 "… *Moreover the cross with independent…*": It is unclear what cross (validation?) refers to, please revise.
CORRECTED. The line 15 has been revised.

Lines 27 – 28 *"…hence the will to focus on aerosol-free regions of the atmosphere with atmospheric signal by molecules from the Rayleigh channel."*: This part of the sentence is unclear, please rephrase.

CORRECTED. The lines 24 – 28 have been rephrased.

Lines 28 – 29 *"…attenuation of the overlying molecular atmosphere…"*: Do the authors mean attenuation of the laser beam due to the overlying molecular atmosphere? Please clarify.

CORRECTED . Yes, the lines 28 – 29 have been rephrased.

Line 38 *"…higher horizontal resolution."*: Maybe provide here the resolution value for the sub-BRC level product?

CORRECTED

Line 48 *". The limited geolocation"*: I think that the word "limited" may give a negative sense, while such small geolocation/time offsets can provide a very good collocation as you mention below. Kindly consider to change the "limited" with another word (e.g. small).

CORRECTED. The lines 48 and 385 have been rephrased.

Line 84 *"…derived from the cross-talk products…"*: Do the authors mean "derived from the cross-talk corrected products"? Kindly revise accordingly.

CORRECTED . Yes, the line 84 has been rephrased.

Line 85 *". They …"*: It is not so clear in which quantities do the authors refer to. To my understanding they are the $\beta_{part}$, $\beta_{mol}$, $\alpha_{part}$. I would suggest to rephrase or maybe add their shortcuts (e.g. $\beta_{part}$, $\alpha_{part}$, etc).

CORRECTED

Lines 96 – 97 *"…signal accumulation of 600 consecutive laser pulses comprised in measurements up to 30, then averaged over ≈ 90 km horizontal to form an observation referred as BRC."*: It may be unclear for a reader not so familiar with the concept of 30 measurements averaged to produce one BRC. I would kindly suggest to rephrase.

CORRECTED . The line 95 – 99 have been rephrased.

Line 98 *"…with measurement level and…"*: The measurement level (i.e. horizontal resolution ~ 18 km) is defined in Line 102 but I think it should be introduced here firstly, mentioning that the measurement level corresponds to 18 km hor. resolution.

CORRECTED . The lines 95 – 99 have been rephrased.

Line 102 *"…It was decided…"*: It was decided from who? The authors of the current study, the MLE developers, the Aeolus DISC team? I would suggest to rephrase and add that the MLE-sub is an additional product next to the original MLE product (in BRC level) and both products are available.
CORRECTED. It was decided from the L2A development team.

Line 103 *"… JATAC September 2022 settings…"*: I don't get what kind of JATAC 2022 settings the authors refer to? Kindly include more details.
CORRECTED. The settings referred to the number of data element « measurement » comprised in 1 BRC (i.e., 5 consecutive measurements per BRC in September 2022). Line 103 has been rephrased in parallel to lines 95 – 99.

Lines 107 – 109 *"The L2A products …. for particles $\alpha_{part}$."*: This sentence repeats what is mentioned in lines 103 – 104 ("…both SCA and MLEsub … referred as measurement level."). But, in lines 107 – 109 the "then" word implies that you apply further processing to the SCA attn. backscatter and the MLE-sub products to align them to the measurement level, which is not correct as the SCA attn. backscatter and the MLE-sub are already provided in measurement level. Please revise accordingly.
CORRECTED. The lines 107 – 109 have been rephrased.

Lines 115 – 122 *"The main objectives… convection (Flamant et al., 2024)."*:  These lines should be rephrased to better explain the JATAC objectives (main objective of Aeolus validation and additional scientific objectives), the lidar instrumentation the authors would like to focus on (maybe split into ground-based and airborne instrumentation, rather than include some of them in one sentence as is now). I think that the Safire Falcon-20 aircraft and the deployed doppler lidar for Aeolus validation should also be mentioned next to the DLR aircraft. Moreover, the discussion about HALO and DC-8 should be moved to the CPEX-CV section. Kindly check for more details on the deployed instruments and observational platforms as well as the JATAC objectives the abstract of Fehr et al.: The Joint Aeolus Tropical Atlantic Campaign 2021/2022 Overview– Atmospheric Science and Satellite Validation in the Tropics, EGU General Assembly 2023, EGU23-7249, https://doi.org/10.5194/egusphere-egu23-7249, 2023.
The lines 115 – 122 have been rephrased. The references for JATAC (i.e., Fehr et al., 2023) and for the high-spectral-resolution Doppler lidar (LNG) (i.e., Bruneau et al., 2015) onboard Safire Falcon-20 aircraft have been added in section 2.2. The Fig. 2 and the discussion about HALO and DC-8 have been moved to section 3.

Line 129 *"CPEX-AW campaign"*:  Are there any references to be cited for CPEX-AW?
Yes ; dedicated webpages are available at https://asdc.larc.nasa.gov/project/CPEX-AW and at https://www.earthdata.nasa.gov/data/projects/cpex-aw. The two references have been added into the section 3.1.

Line 132 – 133 *"On-board instruments and dropsondes have then be operated to be cross analysed with external aerosol retrievals"*: I would suggest to remove the "then" word and rephrase as it erroneously implies that the instrumentation in CPEX has been collecting measurements with the goal to be cross analysed with external instrumentation, which is something that is performed in the context of the current study.

CORRECTED. The corresponding sentence has been removed.

Line 140 – 141 *"It is an active instrument differential absorption lidar (DIAL) and HSRL with multiple configurations including water vapor DIAL and HSRL, and methane DIAL and HSRL."*: Are there any publications where the HALO instrument is described and can be used as reference for more details from the readers?

Yes, the following publications mentioned in introduction and within section 2.2 can be used as references for more details : Bedka et al., 2021 ; Carroll et al., 2022, and Nehrir et al., 2017, 2018. A new reference (i.e., Barton-Grimley et al., 2022) has been added in section 3.2 with the references mentioned above.

Lines 145 – 146 *"…, the spectral dependence of optical properties for desert dust being known to be less pronounced between 355 nm and 532 nm than between 355 nm and 1064 nm"*: Here the authors could add a comment that this is the reason why the HALO products at 532 nm are preferred over the products at 1064 nm for the conversion to 355 nm.

CORRECTED

Lines 147 – 148 *"Moreover HALO transmits linear polarization whereas Aeolus transmits circular polarization"*: Here the authors could add a comment on if and how this difference is being considered for the HALO and Aeolus cross-comparisons.

CORRECTED

Lines 151 – 168: While eq. (1) contains terms that depend on the HALO measuring technique, such as the measured signal and the transmission term for the iodine filter, which is not introduced before and it is unclear for a reader not familiar with the HSRL technique how this filter and its specs affects the retrieval of the extinction, equations 2 – 4 are generic equations. I would suggest to include same level of detail for eq. 2 as for eq. 1 or include the equations for the calculation of the cross-polarized and co-polarized backscatter coefficients. Moreover, the total scattering ratio is introduced in eq. 4 and used in the analysis later, but it is not mentioned before as a parameter that is used in this study (e.g. in lines 148 – 150). As such, kindly revise the section where you describe which HALO parameters are used in the study.

The use of HALO total scattering ratio at 532 nm, HALO linear depolarization ratio at 532 nm, and HALO dominant aerosol type as third variables in two dimentional histograms has been mentioned in section 3.2, and a reference for HALO dominant aerosol type product (i.e., Burton et al., 2012) has been added. The co-authors and I propose to not add further text for equations 1 to 4 as the reference Hair et al., 2008 goes into detail on the entire retrieval.

Equation 1: Does this δ/δr means the partial derivative? The δ symbol is also used for the depolarization ratio. Please clarify and use the symbols that are commonly used for the partial derivative.

Yes, the δ/δr means the partial derivative. It has been corrected with symbols $\partial/\partial r$.

Lines 158 – 159 " *is filtered molecular scattering channel*": It is not clear if this is the measured signal. Kindly clarify in the text.

The sentence has been rephrased with "*is filtered molecular backscattered signal*".

Line 181 *"It is proposed to…"*: From who is it proposed? I would suggest to revise and maybe use a more neutral phrase to start this sentence.

CORRECTED

Line 191 "*…optical properties are cloud-screened and are quality assured…*": Please clarify in the text which optical parameters ( and or Angstrom exponents) and from which instrument (PollyXT or HALO).

CORRECTED

Lines 216 *"A mean value of $5.0^{-4}$ has then been derived and used as a constant for all scenes*": Please add the units and clarify in the text what this constant is (is it the molecular extinction above DC-8 up to 80 km?).

Yes, the constant of $5.0 \times 10^{-4}$ m$^{-1}$ is the molecular extinction above the DC-8 up to 80 km altitude. The unit has been added and the text has been rephrased in section 4.1.

The constant is calculated with Eq. (12) using pressure and temperature information from Numerical Weather Prediction (NWP) model from the European Centre for Medium-Range Weather Forecasts (ECMWF). The figure below illustrates the estimation per ECMWF NWP range bin for Aeolus measurement 1 [13.3° N - 24.3° W ] on 09 September 2022. The sum of the NWP bins from 10 km to 80 km is assigned for the column, and is equal to $5.0 \times 10^{-4}$ m$^{-1}$. The deviations per consecutive column (of order ~ $1.0 \times 10^{-8}$ m$^{-1}$), and from case 09 September to 15 September 2022 and 16 September 2022 (of order ~ $1.0 \times 10^{-7}$ m$^{-1}$) are small and the constant has therefore been added to the $\alpha_{mol,355}$ derived in Eq. (11) when calculating the HALO 355 nm attenuated backscatter coefficient for particles in Eq. (12), and such for all scenes.

[Figure]

Figure. Molecular extinction coefficient (alpha_mol, unit m$^{-1}$ in x axis) derived per ECMWF NWP range bin with altitude (km) in y axis.

Lines 219 – 220 *"Only the valid bins with positive values are considered for statistics (i.e., the invalid measurements coded as NaN in one product being ignored on the second dataset and vice-versa)."*: Please clarify if you apply this flagging (i.e. only positive values) for all the products from Aeolus and HALO that are used in the cross-comparisons.
This flagging is applied for all the products from Aeolus and HALO used in the cross-comparisons. The lines 219 – 220 have been corrected.

Lines 231 – 234 *"The HALO 532 channel… 300 m vertical average."*: I find hard to easily understand the difference on the vertical resolution (i.e. bin length of 30 m) and the sampling interval. Does this sampling interval correspond to the ~0.5 seconds of sampling? Please revise this section and clarify what is the vertical resolution and the sampling interval. Moreover, if the vertical resolution is 30 m, then why do you interpolate to 15 m, please clarify. In line 233, please name, next to the extinction coefficient, which products that are used in the study are calculated after further vertical averaging of 300 m.
The HALO 532 nm channels are sampled at 0.5 temporal and 1.25 m vertical resolutions. The corresponding paragraph in section 4.2 has been revised.

Lines 234 – 235 "The HALO data are sampled each ≈ 0.5 seconds, *a 10 second average being applied to the backscatter coefficient along the direction of flight*": Does this mean that only the backscatter coeff. is calculated after an additional 10-second averaging is applied? What about the rest products that are used in the study. Please clarify and revise the text if needed.
The paragraph in lines 231 - 235 has been revised.

Figure 3 "*The NASA DC-8 flight track is superimposed in red color code and the 6 consecutive BRC level observations of Aeolus orbit file no. 23562 are displayed (a). They correspond to 30 Aeolus measurements given at sub-BRC level ≈ 18 km, and to 225 HALO profiles (b)*": I find these two sentences a bit confusing so I would kindly suggest to split the info (flight track and number of profiles) in two separate parts, one for HALO and one for Aeolus.
CORRECTED

Line 259 – 260 "*Figure 4a illustrates how the Aeolus MLEsub βpart up to 18 km altitude helps assessing the particle-free conditions above the DC-8 flying at ≈ 10.7 km altitude for the 16 September 2022 case*": Please add a brief discussion on the "how", bridging with lines 251 – 254.
CORRECTED. The paragraph has been modified, pointing to the dominant dark blue color code for close-to-zero backscatter above 10 km altitude.

Lines 261 – 263 "*PollyXT emitting linear polarization (Engelmann et al., 2016), the circular depolarization ratio at 355 nm is derived to recompute the parallel backscatter coefficient at 355 nm with equations similar to Eqs. (5) and (6)*": I find this sentence confusing, kindly re-phrase.
CORRECTED

Line 265 "*Figure 4b shows how the HALO 532 nm…*": Please add that the figure shows also the converted PollyXT 355 parallel backscatter.
CORRECTED

Line 266 "*The HALO parallel 355 nm profile agrees with Aeolus MLEsub (i.e., violet)*": Since the authors denote the color of the Aeolus MLEsub backscatter, consider adding the color info also for the HALO parallel 355 backscatter. Moreover, I would suggest to discuss also the underestimation of MLEsub at 3 km.
The Angstrom exponents' median values of the daily profiles for 09, 15 and 16 September 2022 have been derived from the PollyXT measurements instead of the monthly profiles and used to reprocess the 532 nm to 355 nm conversion of the HALO atmospheric products. A new co-author who derived the Ansgtrom exponents values was added (i.e., Moritz Haarig from TROPOS). This results in a much better agreement between the MLEsub and HALO parallel 355 nm backscatter profiles in Figs. 4b below, A4b and A5b; the HALO and PollyXT parallel 355 nm backscatter profiles being now really close for the inner SAL at 3 km altitude.

[Figure]

Figure 4b. Parallel backscatter coefficient for particles at 355 nm derived with Aeolus MLEsub algorithm (i.e., violet), PollyXT (i.e., blue) and HALO (i.e. dark yellow) on 16 September 2022

Line 273 – 275 "*Similar observations can be made with case 09 September… (Appendix A4b)*": Although the case on 15 September is mentioned as one of the three cases, only the results for 9 and 16 September are discussed. Please add a dedicated discussion also for the results on 15 Sep. CORRECTED

Figure 4: (4b) The converted HALO parallel backscatter 355 nm does not agree with the parallel backscatter from PollyXT, but agrees with the total backscatter from PollyXT inside the dust layer heights (1-4 km). This difference is also visible at 2 km in Figure A4(b), but in Figure A5(b) the profiles are as expected (i.e. similar values for the parallel backscatter 355 from PollyXT and HALO). Could it be the depolarization ratio from the two systems that is used in the conversions, or the backscatter related Angstrom exponent? Maybe an additional subplot with the particle linear depolarization ratio profiles (355 nm for PollyXT; 532 nm for HALO) that are used in the conversions could add in the discussion on the observed differences? As such, I think that a dedicated part where the authors discuss these differences and with their potential source(s) should be added in section 5.1. Moreover, based on the presented results and the discussion, a concluding remark should be added on the evaluation of the wavelength and total to parallel conversions.
The Angstrom exponents' median values of the daily profiles for 09, 15 and 16 September 2022 have been derived from the PollyXT measurements instead of the monthly profiles for the preprint. This results in a much better agreement between the MLEsub, HALO and PollyXT parallel 355 nm backscatter profiles (Fig. 4b above within reply on comment for line 266). Nevertheless, some deviations remains and the HALO parallel 355 nm backscatter (Fig. 4b, dark yellow) generally looks smaller than the PollyXT 355 nm backscatter (Fig. 4b, blue). A closer look at the HALO and PollyXT linear and circular aerosol depolarization ratio reveals offset between the products measured at 532 nm, i.e., the HALO aerosol depolarization ratios at 532 nm (Figure below, red) being overestimated when compared to the PollyXT (Figure below, blue). This is still under investigation as the calibration of the two instruments must be in cause. Both instruments show coherent values of particle linear depolarization ratio at 532 nm for Saharan dust at 3 km altitude (i.e. values of order ~ 0.25 to 0.30 for SAL in accordance with literature such as Floutsi et al., 2023).

[Figure]

Figure. Linear aerosol depolarization ratios and circular aerosol depolarization ratios derived from PollyXT measurements at 355 nm and 532 nm, and from HALO measurements at 532 nm for the closest profiles to Mindelo, Cabo Verde on 09, 15 and 16 September 2022.

(4c) It seems to me that when comparing the extinction from HALO and PollyXT, a small vertical displacement is visible (e.g. check the peak at ~2.6 km for HALO instead the peak at ~3.5 km for PollyXT). Could the authors comment on this?
The offset between HALO and PollyXT peaks for extinction coefficient (Fig. 4c below) may be due to the different calibration between HALO and PollyXT in addition to the geolocation offset (i.e. of ~ 11.5 km) and different geometry (off-nadir ~ 5° from ground for PollyXT and nadir viewing from top for HALO). Figs. 4a and 9 show a SAL vertically extended from 2 km to 5 km altitude and the two instruments may have indeed measured optical properties of two differents filaments of the SAL.

[Figure]

Figure 4c. Extinction coefficient for particles at 355 nm derived with Aeolus MLEsub algorithm (i.e., violet), PollyXT (i.e., blue) and HALO (i.e. dark yellow) on 16 September 2022.

Line 299 "…*for second-last measurement 34 is even more clear.*": I would suggest to specifically mention to which instrument/platform (Aeolus or HALO) belongs the mentioned measurement number (e.g. "for second-last Aeolus measurement 34"). Kindly harmonize throughout the whole section.
CORRECTED

Line 314 "*up to ≈ 3$^{-3}$ km$^{-1}$ sr$^{-1}$*": Do the authors mean $3 \cdot 10^{-3}$? If yes, kindly correct it and harmonize all similar fields (e.g. line 316) with the correct format.
Yes, the format $3.0 \times 10^{-3}$ has been selected for all similar fields.

Line 316 "…*with values below ≈ 2 $^{-3}$ km$^{-1}$ sr$^{-1}$*…": please clarify if the mentioned values are measured from Aeolus of HALO.
CORRECTED

Line 323 *"The HALO dominant aerosol type points to marine aerosols"*: The third regime in Figure 6f contains also points classified as dusty mix, so why do the authors claim that the dominant type is marine aerosols? Could the authors support their claim (e.g. number of points classified as marine vs dusty mix) and include a dedicated part in the text?
CORRECTED. The sentence has been rephrased with *"The HALO dominant aerosol type points to marine aerosols and dusty mix"*.

Figure 6: A general suggestion for the HALO particle depolarization ratio that is illustrated in all the 2-D histograms and discussed in the subsections of 5.2 is to use the particle linear depolarization ratio that is directly retrieved from HALO, instead of the converted circular depolarization ratio, in the corresponding figures and in their discussion throughout the manuscript as the linear depolarization ratio can be easily linked with references to past studies for aerosol characterization (e.g. Freudenthaler et al., 2009). However, the authors may also keep (as an additional info) the indicative particle circular depolarization ratio values that are already mentioned in the discussion of the related figures.
CORRECTED. The Figs. 6a-c, 8a-c, 10a-c, A6, A7, A8 have been reprocessed with particle linear depolarization ratio as third variable. The indicative particle circular depolarization ratio values are mentioned in the discussion.

Lines 339 – 340 *"…deviations correspond to the first regime where aerosol loading is weak and falls below the classification threshold of 0.2 in HALO aerosol scattering ratio at 532 nm $\sigma_{aer,532}$"*: I would suggest adding the color code that corresponds to the first regime too. Moreover, the threshold value of $\sigma_{aer,532} = 0.2$ contradicts with the definition of $\sigma_{aer,532}$ in Eq. 4 which cannot take values less than 1. Please revise Eq. 4 and clarify throughout the manuscript which one of the total/aerosol scattering ratio is used.
The data for low scattering have a dedicated color codes (i.e. gray in Figs. 6d-f, 8d-f, 10d-f, and cyan in Appendix A6, A7, A8). The Eq. (4) should correspond to HALO total scattering ratio at 532 nm, and not HALO aerosol scattering ratio at 532 nm. The Eq. (4) and the paragraph in section 5.2.2 have been revised. The ready-to-use HALO products use a classification threshold of >1.2 in HALO total scattering ratio at 532 nm for calculating intensives like depolarization. The symbol $\sigma_{aer,532}$ has been replaced with $R_{532}$ for HALO total scattering ratio at 532 nm.

Lines 347 – 348 *"Removing the contributions from low scattering below HALO $\sigma_{aer,532}$ of 1.1 has a significant impact on the slope (i.e., increased from 0.29 to 0.48)"*: The authors apply this threshold only in the comparison for the SCA attenuated backscatter. It would be interesting to see the scores after applying the threshold also in the MLEsub comparisons.
The application of the HALO total scattering ratio $R_{532}$ threshold of 1.2 results in increase of slope for MLEsub particulate backscatter (i.e., from 0.71 to 0.91).

Lines 372 *"One could flag the cloudy regions applying a scattering ratio threshold of 5…"*: Same as previous comment (Lines 347 – 348). It would be interesting to see if this total scattering ratio mask ($1.1 < \sigma_{aer,532} <= 5$) for low scattering and cloud contaminated regimes would improve the comparison between Aeolus and HALO also in the cases on 9 and 15 September. Could the authors support why do they apply this scattering ratio mask only for the case on 16 September, while clouds are also present in the cross sections of 9 and 15 September?
The scattering ratio mask is applied for the 16 September 2022 as it corresponds to the lower metrics R² and RMSE in 2D histograms with biggest outliers (Fig. 10), and because it is the only case whith dense clouds above 5 km altitude which may attenuate the signal, leading to lower SNR in low altitudes. It was assumed that the scattering ratio mask would have a more prominent impact in such conditions. The mask results in decrease of RMSE for SCA particulate attenuated backscatter on 16 September 2022 (i.e. from 0.97 to 0.71).
The mask has been applied to the other cases for testing purpose and a positive impact is also reported for MLEsub on 09 September 2022 (i.e. decrease of RMSE from 0.59 to 0.42 for backscatter and from 0.68 to 0.36 for extinction) and on 15 September 2022 (i.e. decrease of RMSE from 0.59 to 0.46 for backscatter and from 0.66 to 0.64 for extinction).

Line 275 *"Marine aerosols are well characterized by Aeolus SCA (blue color code in Fig. 10f)"*: Is Aeolus SCA a typo here? Do the authors actually refer to the characterization of marine aerosols by HALO? Please clarify and revise the sentence.
CORRECTED. The sentecne has been rephrased as the main message here is to emphasize the signal underestimation by the SCA algorithm for the marine aerosol regime indicated with blue color code in Fig. 10f.

Lines 411 – 414 *"We then do recommend … then appearing as the best compromise"*: In the results section I didn't see any discussion about the different range bin settings and how they impact the scores, as the regimes are identified by the depolarization ratio or the aerosol type products from HALO. I would suggest the authors to include relevant discussion about the impact of range bin settings in the "Results" section before concluding to the recommendations in the "Conclusions" section.
The lines 411 – 414 have been rephrased. The Figs. 5, 7 and 9 shows that the MLEsub quality flagging based on SNR and error estimates is more restrictive in low altitudes below 2 km, then corresponding to Aeolus range bin thickness of 500 m (i.e. contrary to 750 m thickness from 2 km to 9 km altitude and 1000 m thickness above 9 km altitude, for the Cabo Verde region and in September 2022).

**Technical corrections**

Line 4 *"… (i.e., backscatter, extinction, attenuated backscatter, and depolarization coefficients) …"*: Suggested change to "backscatter, extinction, attenuated backscatter coefficients, and depolarization ratios"?
CORRECTED

Line 29 *"…decreases with altitude…"*: Suggested change to " decreases with altitude decrease"?
CORRECTED

Line 29 *"…the smaller values…"*: Suggested change to " with the smaller SNR values"?
CORRECTED

Line 31 *"forms an L2A"*: "forms a L2A"
CORRECTED

Line 49 *"…of these flights with Aeolus…"*: Suggested change to "between DC-8 flights and Aeolus"
CORRECTED

Line 91 *". It makes use …"*: Suggested change to "The MLE approach makes use"
CORRECTED

Line 218 *"… as SCA particulate attenuated backscatter…"*: is something missing here?
CORRECTED ; rephrased as *", and SCA particulate attenuated backscatter coefficient,"*.

Line 265 *"before crossing Aeolus and HALO…"*: do the authors mean before comparing (or cross analyzing) Aeolus and HALO?
CORRECTED ; rephrased as *"before comparing Aeolus and HALO"*.